# The S-phase-induced lncRNA *SUNO1* promotes cell proliferation by controlling YAP1/Hippo signaling pathway

Qinyu Hao[1†], Xinying Zong[1†], Qinyu Sun[1], Yo-Chuen Lin[1], You Jin Song[1], Seyedsasan Hashemikhabir[2], Rosaline YC Hsu[1], Mohammad Kamran[1], Ritu Chaudhary[3], Vidisha Tripathi[1], Deepak Kumar Singh[1], Arindam Chakraborty[1], Xiao Ling Li[3], Yoon Jung Kim[4‡], Arturo V Orjalo[5§], Maria Polycarpou-Schwarz[6], Branden S Moriarity[7], Lisa M Jenkins[8], Hans E Johansson[5], Yuelin J Zhu[9], Sven Diederichs[6,10], Anindya Bagchi[11], Tae Hoon Kim[4], Sarath C Janga[2], Ashish Lal[3], Supriya G Prasanth[1], Kannanganattu V Prasanth[1]*

[1]Department of Cell and Developmental Biology, Cancer center at Illinois, University of Illinois at Urbana-Champaign, Urbana, United States; [2]Department of BioHealth Informatics, School of Informatics and Computing, IUPUI, Indianapolis, United States; [3]Regulatory RNAs and Cancer Section, Genetics Branch, Center for Cancer Research, National Cancer Institute, Bethesda, United States; [4]Department of Biological Sciences and Center for Systems Biology, The University of Texas at Dallas, Richardson, United States; [5]LGC Biosearch Technologies, Petaluma, United States; [6]Division of RNA Biology and Cancer, German Cancer Research Center (DKFZ), Heidelberg, Germany; [7]Department of Pediatrics, University of Minnesota, Minneapolis, United States; [8]Center for Cancer Research National Cancer Institute, Bethesda, United States; [9]Molecular Genetics Section, Genetics Branch, Center for Cancer Research, National Cancer Institute, Bethesda, United States; [10]Division of Cancer University of Freiburg, German Cancer Consortium (DKTK), Freiburg, Germany; [11]Sanford Burnham Prebys Medical Discovery Institute, La Jolla, United States

*For correspondence:
kumarp@illinois.edu

[†]These authors contributed equally to this work

Present address: [‡]Children's Research Institute (CRI), UT Southwestern Medical Center, Dallas, United States; [§]Genentech Inc, South San Francisco, United States

**Abstract** Cell cycle is a cellular process that is subject to stringent control. In contrast to the wealth of knowledge of proteins controlling the cell cycle, very little is known about the molecular role of lncRNAs (long noncoding RNAs) in cell-cycle progression. By performing genome-wide transcriptome analyses in cell-cycle-synchronized cells, we observed cell-cycle phase-specific induction of >2000 lncRNAs. Further, we demonstrate that an S-phase-upregulated lncRNA, *SUNO1*, facilitates cell-cycle progression by promoting YAP1-mediated gene expression. *SUNO1* facilitates the cell-cycle-specific transcription of *WTIP*, a positive regulator of YAP1, by promoting the co-activator, DDX5-mediated stabilization of RNA polymerase II on chromatin. Finally, elevated *SUNO1* levels are associated with poor cancer prognosis and tumorigenicity, implying its pro-survival role. Thus, we demonstrate the role of a S-phase up-regulated lncRNA in cell-cycle progression *via* modulating the expression of genes controlling cell proliferation.

## Introduction

Cell-cycle progression is a vital cellular process, subject to stringent control, as aberrant cell-cycle progression usually results in genome instability, contributing to cancer progression (*Robertson et al., 1990*; *Cho et al., 2001*; *Dyson, 1998*; *Frolov and Dyson, 2004*; *Sánchez and*

*Dynlacht, 1996*). The eukaryotic cell cycle is controlled by a regulatory network, which proceeds through tightly regulated transitions to make sure that specific events occur in an orderly fashion. The activity of genes that control cell proliferation is strictly regulated through the cell-cycle-dependent oscillation of their expression (*Robertson et al., 1990*; *Cho et al., 2001*; *Dyson, 1998*; *Frolov and Dyson, 2004*; *Sánchez and Dynlacht, 1996*). Such dynamic changes in gene expression during cell cycle are essential for efficient cell-cycle progression (*Robertson et al., 1990*; *Cho et al., 2001*; *Dyson, 1998*; *Frolov and Dyson, 2004*; *Sánchez and Dynlacht, 1996*). For example, studies have established the role of transcription factors (TFs) such as the E2F and TEAD family of proteins in regulating the transcription of genes controlling cell cycle and cell proliferation (*Frolov and Dyson, 2004*; *Chen et al., 2009*; *Harbour and Dean, 2000*; *Meng et al., 2016*). Extensive studies on the identification of protein-coding genes exhibiting periodic expression patterns during cell cycle have led to improved understanding of the basic cell-cycle process and its regulatory mechanism, exemplified by studies on cyclins (*Pines and Hunter, 1989*). Understanding the mode of cell cycle-regulated gene expression is also central to the study of many diseases, most prominently cancer. Thus, characterization of the genome-wide changes in the transcriptional program during the cell cycle is a critical step toward a deeper mechanistic understanding of the cell proliferation process and its role in cancer.

One of the most unexpected discoveries in the genomics era of biology is the extensive transcription of RNA from non-protein-coding regions of the genome (www.gencodegenes.org). Tens of thousands of long noncoding RNAs (lncRNAs), defined as transcripts larger than 200 nt with no or low protein-coding potential, have been identified in mammalian cells. Pioneering studies on a small proportion of lncRNAs revealed that lncRNAs are an integral part of the cellular control network that co-exists along with proteins (*Goff and Rinn, 2015*; *Yao et al., 2019*; *Kopp and Mendell, 2018*; *Sun et al., 2018a*; *Quinn and Chang, 2016*; *Rinn and Chang, 2012*) and play important roles in cancer (*Gutschner et al., 2013*). Mechanistically, the RNA sequence and structure offer lncRNAs two inherent functional properties: (1) sequence-mediated interaction with genomic DNA or other RNA, and (2) secondary/tertiary structure-mediated interaction with RNA-binding proteins. With these properties, lncRNAs modulate the recruitment of TFs, cofactors or chromatin modifiers to specific genomic locus, to regulate gene expression transcriptionally or epigenetically; or to regulate the binding of RNA processing factors or microRNAs to pre-mRNAs or mRNAs, thereby influencing gene expression at the post-transcriptional level (*Batista and Chang, 2013*). Functionally, lncRNAs control several biological functions, including but not limited to processes such as dosage compensation, genomic imprinting, cell metabolism, differentiation and stem cell pluripotency (*Goff and Rinn, 2015*; *Kopp and Mendell, 2018*; *Sun et al., 2018a*; *Quinn and Chang, 2016*; *Rinn and Chang, 2012*).

In contrast to the wealth of knowledge of proteins involved in the regulation of the cell cycle, and associated with oncogenic mutations, very little is known about the molecular role of cell-cycle phase-regulated lncRNAs. Recent studies have indicated that several lncRNAs regulate vital biological processes such as cell cycle, cell proliferation and DNA-damage response, *via* either directly regulating DNA replication or indirectly controlling the expression of critical cell-cycle regulatory genes (*Schmitt and Chang, 2016*; *Li et al., 2016*; *Kitagawa et al., 2013*). Examples include Y RNA, which is involved in the activation of replication initiation (*Kowalski and Krude, 2015*), *MALAT1* that promotes the expression and activity of TFs such as E2F and B-Myb (*Tripathi et al., 2013*; *Ji et al., 2003*), and the recently reported *CONCR*, a lncRNA whose expression is periodic during cell cycle, controls sister chromatid cohesion by regulating the activity of DDX11 helicase (*Marchese et al., 2016*). In addition, LncRNAs such as *p15-AS, lincRNA-p21, RoR, PANDA, DINO* and *NORAD* are known to regulate cell-cycle progression through modulating the tumor-suppressor and growth-arrest pathways during senescence and in response to DNA damage (*Petermann et al., 2010*; *Zhang et al., 2013*; *Schmitt et al., 2016*; *Lee et al., 2016*). Also, elegant studies have demonstrated that a subset of lncRNAs transcribed from or near the promoters of cell-cycle-regulated protein-coding genes were shown to have coordinated transcription with their respective protein-coding genes, in response to diverse perturbations, including oncogenic stimuli, stem cell differentiation or DNA damage, suggesting their potential biological functions (*Schmitt et al., 2016*; *Hung et al., 2011*; *Goyal et al., 2017*). Finally, by performing CRISPR/Cas9- or CRISPRi-mediated of depletion of >1000 s of lncRNAs in multiple cancer cell lines, a recent study had reported that ~ 100 lncRNAs regulate cell growth and cell viability in a cell type-specific manner, though the molecular function of

these lncRNAs is yet to be determined (*Liu et al., 2017a*). Despite these studies, our understanding on the mechanistic role of lncRNAs during cell-cycle progression remains extremely limited. A comprehensive characterization of the expression of lncRNAs during cell cycle would generate a rich resource for further characterizing lncRNA-mediated regulatory networks, contributing to cell-cycle progression. In addition, such a dataset would provide insights into how lncRNAs are exploited by tumorigenic mutations that drive malignancy.

Here, we systematically profiled the expression of both protein-coding and lncRNA genes during cell cycle by performing deep RNA-seq of cell-cycle-synchronized (G1, G1/S, S, G2 and M-phases) cancer cells, and identified >2000 lncRNAs that displayed periodic expression, peaking during specific phases of the cell cycle. Mechanistic studies on a S-phase-upregulated novel lncRNA that we named as *SUNO1* (*S*-phase-*U*pregulated *NO*n-coding-*1*) revealed its vital role in modulating the Hippo/Yap1 signaling pathway, thereby promoting cell-cycle progression.

## Results

### Transcriptome analyses of cell-cycle-synchronized cells reveal cell-cycle-regulated expression of protein-coding and noncoding genes

To determine non-random cyclical changes in gene expression during cell-cycle progression, we performed paired-end deep RNA-sequencing (>100 million paired-end reads/sample) of the osteosarcoma cells U2OS that were synchronized into discrete cell-cycle stages: G1, G1/S, S, G2 and M (please see Materials and methods for synchronization details). (*Figure 1A* and *Figure 1—figure supplement 1A*). Principal component analysis confirmed that the data set from biological replicates was highly consistent in our RNA-seq data sets (*Figure 1—figure supplement 1B*). U2OS cells showed quantifiable expression (CPM $\geq$ 0.075 in at least two samples) of ~24,087 genes, including 15,780 coding and 8307 non-coding genes, including 7836 potential lncRNAs (*Figure 1—figure supplement 1C*; *Supplementary file 1*). Transcriptome profiling revealed dynamic expression of genes during cell-cycle progression (*Figure 1—figure supplement 1C*). In order to assess the biological processes/pathways that are activated/repressed during cell-cycle transition, we performed differential expression analyses between two adjacent cell-cycle stages (for example, G1 to G1/S or G1/S to S) (*Figure 1B*; *Supplementary files 2* and *3*). In this case, we defined differentially expressed genes (DEGs) as genes that displayed |fold change| $\geq$ 1.5 and FDR < 0.05, in statistical analysis. We observed differential expression of several thousands of genes during cell-cycle stage transition (10984 DEGs between G1 to G1/S; 5117 DEGs between G1/S to S; 3947 DEGs between S to G2; 10586 DEGs between G2 to M; and 8229 DEGs between M to G1), including the established cell-cycle regulators such as *cyclins* (*Figure 1B* and *Figure 1—figure supplement 1D*; *Supplementary file 3*). Interestingly, we observed that ~ 35–40% of the genes that showed differential expression during a particular cell-cycle stage transition consisted of lncRNAs (3529 in G1 to G1/S; 2195 in G1/S to S; 1553 in S to G2; 3405 in G2 to M and 3074 in M to G1 transition) (*Figure 1B*; *Supplementary file 3*), implying potential roles played by thousands of lncRNAs during cell-cycle progression.

Next, we performed bioinformatic analyses to gain insights into the biological pathways that were associated with the DEGs during cell-cycle stage transition. GSEA analyses revealed that pro-proliferative and oncogenic pathways, such as positive regulators of MAPK cascade were activated during G1/S to S-phase transition (*Figure 1—figure supplement 2A*; *Supplementary file 4*). Pathway analyses indicated that DEGs during G1/S to S transition were enriched for biological processes that promote cancer progression, including the MAPK, RAS and Hippo signaling pathways (*Figure 1—figure supplement 2B*; *Supplementary file 4*), implying an intimate link between differential expression of genes during G1/S to S transition and cancer.

In order to determine if a particular gene participates in a cellular function during a specific cell-cycle phase, we further identified the cell-cycle phase-specific expressed genes from the DEGs described above, by utilizing the following criteria: The genes showing (1) the highest expression in one particular cell-cycle stage compared to rest of the cell-cycle stages, and (2) significantly (FDR < 0.05) higher expression (|Fold change| $\geq$ 1.5) in a particular cell-cycle phase compared to adjacent cell-cycle phases. By this approach, we identified 5162 genes (1409 genes in G1, 1486 genes in G1/S, 575 genes in S, 666 genes in G2, and 1026 genes during M phase) that display

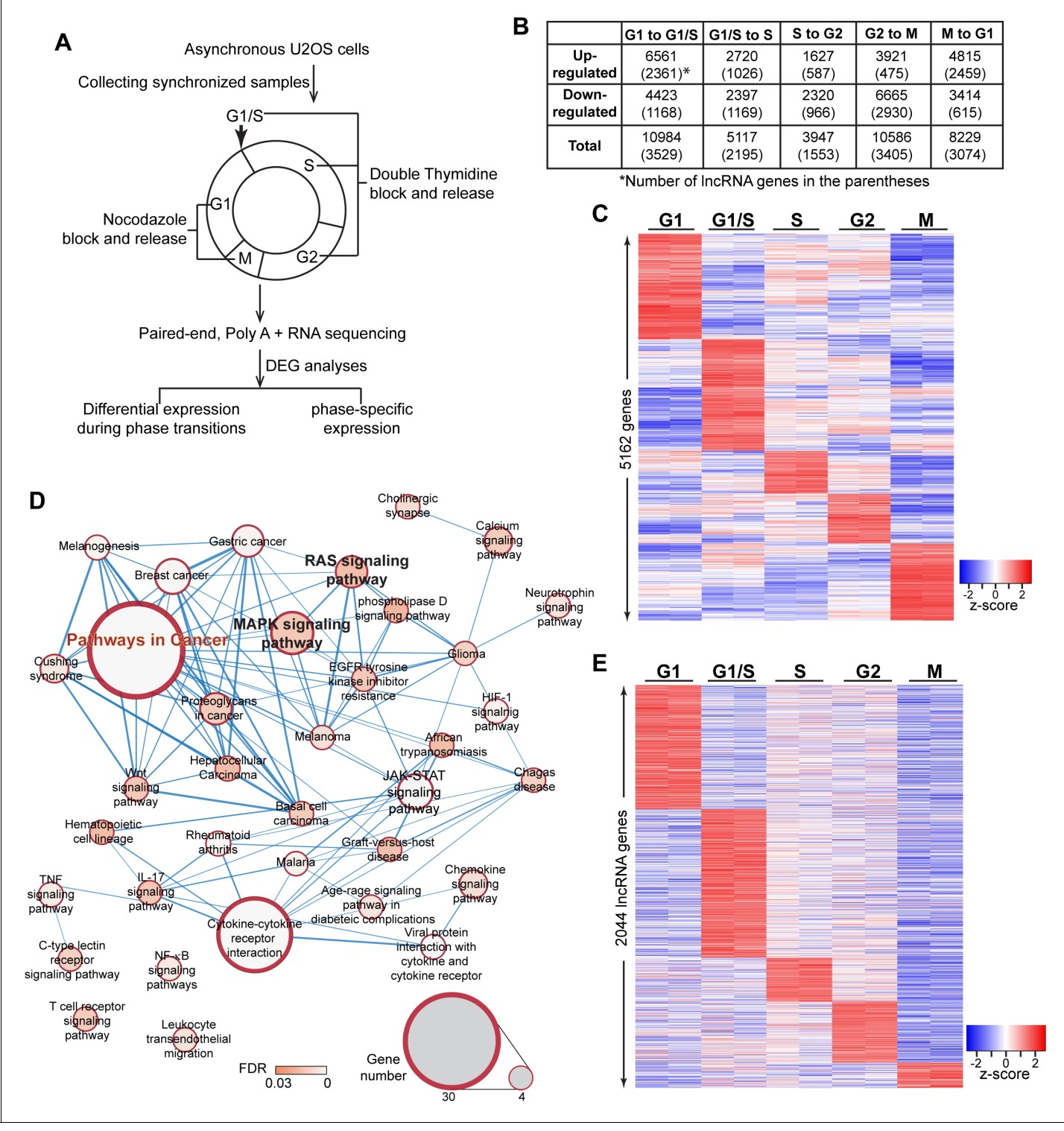

**Figure 1.** Transcriptome landscape of U2OS cells during cell-cycle progression. (**A**) Schematic of sample preparation and analyses pipeline of RNA-seq. U2OS cells are synchronized to different phases of cell cycle (G1, G1/S, S, G2, M) in biological replicates, then subject to paired-end, polyA+, and high depth RNA-seq. Differential expression analyses are performed using gene count data to identify differentially expressed genes comparing every two adjacent phases. Phase-specific genes are further defined as detailly described in Materials and method. (**B**) Table representing the number of differentially expressed genes (DEGs) between every two adjacent cell-cycle phases. The number in the parenthesis refers to long non-coding DEGs. Detailed DEG information is available in *Supplementary file 3*. (**C**) Heatmap of all phase-specific genes. Full list of all 5162 phase-specific genes are

*Figure 1 continued on next page*

Figure 1 continued

listed in *Supplementary file 5*. (D) Top events from Kegg pathway analysis of S-phase-specific genes. Full results are listed in *Supplementary file 4*. (E) Heatmap showing cell-cycle phase-specific expression of lncRNAs in U2OS cells.

The online version of this article includes the following figure supplement(s) for figure 1:

**Figure supplement 1.** Cell-cycle-specific expression of genes in U2OS cells.

**Figure supplement 2.** Pathways and biological processes of genes that showed differential expression during cell cycle.

phase-specific expression (*Figure 1C*; also see *Figure 2—figure supplement 1A* and *Supplementary file 5*). Pathway and Gene ontology analyses revealed important functions attributed to the phase-specifically expressed genes. For instance, S-phase-specific genes participated in several pro-proliferation and cancer promoting pathways, (*Figure 1D*, *Supplementary file 4*). Similarly, M-phase-expressed genes are detected to be relevant to mitotic cell-related biological processes (*Figure 1—figure supplement 2C*, *Supplementary file 4*).

## Transcriptome analyses revealed phase-specific expression of lncRNAs during cell cycle

At present, little is known about the role of lncRNAs that show enhanced expression during a particular cell-cycle phase. We demonstrated a microRNA-independent role for the G1 phase-enriched MIR100 host gene lncRNA in G1/S transition by modulating HuR-mediated mRNA stability and/or translation (*Sun et al., 2018b*). Recently, we reported that *MIR222HG* lncRNA promoted the cell cycle re-entry post quiescence by modulating ILF3/2 activity, further supporting the role of lncRNAs in cell-cycle progression (*Sun et al., 2020*). Our RNA-seq analyses revealed that ~ 42% (2158 out of 5162 genes) of genes that show elevated expression during a particular cell-cycle phase categorized into non-coding genes, in which the majority of them (2044 out of 2158) belonged to one of the several classes of lncRNA genes (630 in G1; 754 in G1/S; 222 in S; 310 in G2, and 128 during M phase) (*Figure 1E* and *Figure 2—figure supplement 1A*; *Supplementary file 5*).

In order to test the hypothesis that similar to protein-coding genes, cell-cycle phase-specific-expressed lncRNAs perform vital roles during cell proliferation, we focused on characterizing the function of an S-phase-enriched lncRNA. By performing RT-qPCR we validated the RNA-seq data, showing S-phase-specific elevated expression of several candidate lncRNAs (*Figure 2—figure supplement 1B*). Further, we performed mechanistic studies to determine the role of one of the novel S-phase-upregulated lncRNAs, *SUNO1* (*S*-phase-*U*pregulated *NO*n-coding-*1*) (AC008556.1; ENSG00000277013; LNCipedia gene ID: lnc-KCTD15-2) (*Figure 2A*) in cell-cycle progression. RNA-seq analyses revealed elevated levels of *SUNO1* in U2OS cells that were synchronized into S-phase (*Supplementary files 3* and *5*). RT-qPCR analyses of cell-cycle-synchronized cells also indicated that *SUNO1* showed elevated levels during G1/S and early stages of S-phase (*Figure 2B* and *Figure 2—figure supplement 1C*). The *SUNO1* gene is a long intergenic lncRNA (lincRNA), located on human Chromosome 19. Analyses, including RT-qPCR, GRO-seq and EU-pulse labeling followed by nascent RNA-seq in various human cell lines (HeLa, MCF7 [*Liu et al., 2017b*] and hTERT-RPE1 [*Yildirim et al., 2020*]) revealed G1/S- and/or S-phase induced expression of *SUNO1*, implying that cell-cycle phase-specific expression of *SUNO1* is not unique to a particular cell line (*Figure 2—figure supplement 1D–F*). RNA-seq data from nine ENCODE human cell lines (*Figure 2A*) as well as RT-qPCR in other human cell lines revealed cell line-specific expression of *SUNO1* (*Figure 2—figure supplement 1G*). For example, the tumorigenic human breast cancer cell line BT-20 showed the highest expression of *SUNO1*, whereas HCT116 and U2OS showed moderate levels of *SUNO1* compared to other cell lines, such as the WI-38 human diploid fibroblasts (*Figure 2—figure supplement 1G*). We utilized HCT116 (pseudo-diploid) and U2OS (aneuploid) cell lines for most of the downstream functional studies. A genome-wide histone-tail modification map indicated significant H3K4me3 and H3K27ac marks on the promoter of *SUNO1* in multiple cell lines (*Figure 2A*). *SUNO1*'s 5'end, including the promoter is located within a long stretch of a CpG island (>1 kb), a scenario normally observed for house-keeping genes (*Figure 2A*). The *SUNO1* sequence seems to be conserved only in primate lineage, except for sequence elements within the 5'end of the gene, including the promoter, which was reasonably conserved among vertebrates (*Figure 2A*). Further, the *SUNO1* promoter showed significant enrichment of RNA pol II (POL2RA) and TFs, such as FOS,

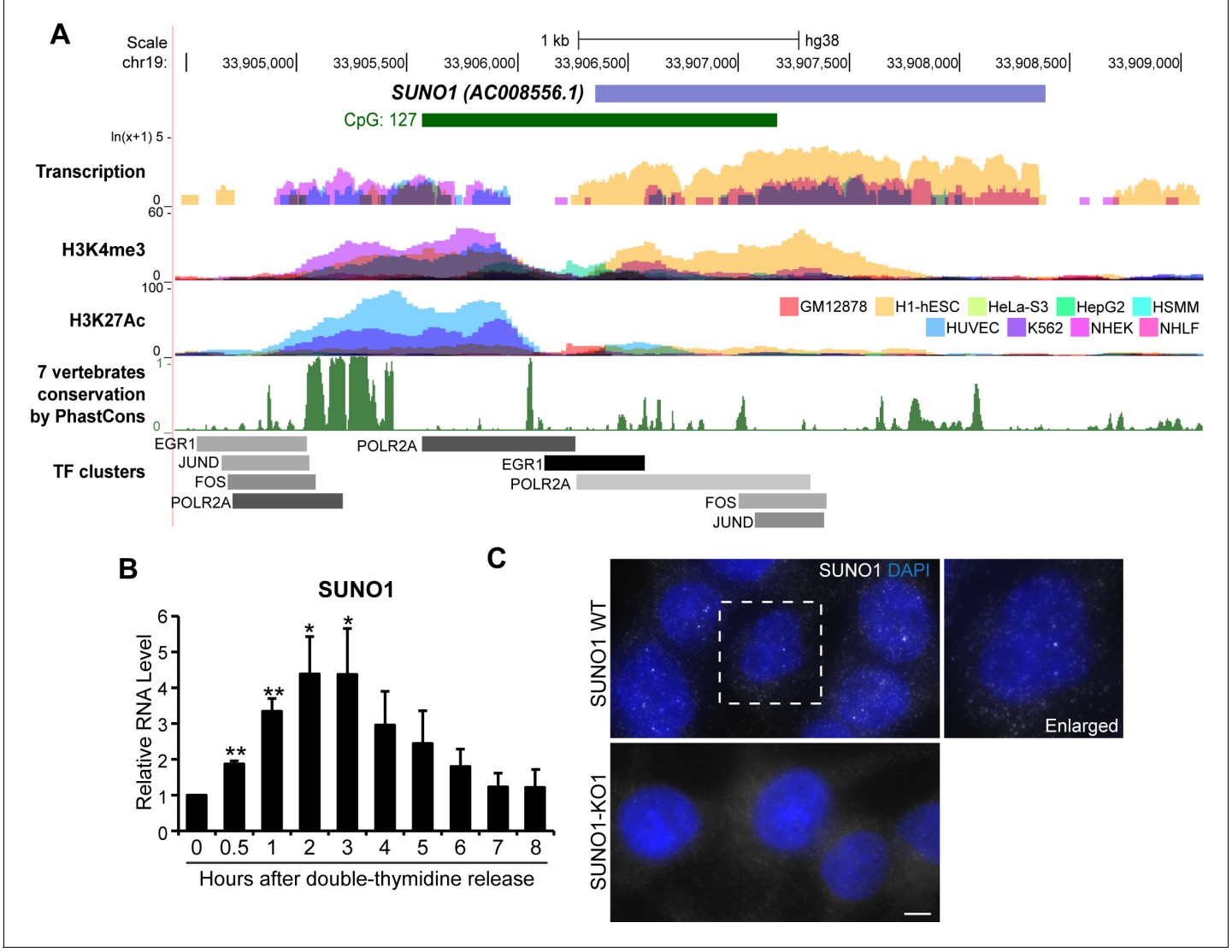

**Figure 2.** *SUNO1* is an S-phase-induced lncRNA. (**A**) UCSC genomic browser view of *SUNO1* genomic locus, showing position of CpG island, transcription in 9 ENCODE cell lines, H3K4me3 ChIP-seq data set, H327Ac ChIP-seq data set, vertebrate conservation, and clusters of Pol II and cell-cycle-regulating transcription factors (TFs) from ENCODE data sets. (**B**) RT-qPCR to detect relative levels of *SUNO1* in U2OS cells post double-thymidine block for indicated time points (hours). Data are presented as Mean ± SD, n = 2. Unpaired two-tail t-tests are performed. *p<0.05, **p<0.01, ***p<0.001. (**C**) Single-molecule RNA-FISH (smRNA-FISH) to detect *SUNO1* RNA in wild-type and *SUNO1* knock-out HCT116 cells. *SUNO1* KO1 cells used as a negative control for *SUNO1* RNA smRNA-FISH. DNA is counterstained with DAPI. Scale bar: 5 μm.

The online version of this article includes the following figure supplement(s) for figure 2:

**Figure supplement 1.** *SUNO1* is upregulated during S-phase.

**Figure supplement 2.** Basic characterization of *SUNO1*.

**Figure supplement 3.** Basic characterization of *SUNO1*.

JUND and EGR1, which are known to induce the expression of genes promoting cell-cycle progression (*Figure 2A* and *Figure 2—figure supplement 2A*).

Cellular fractionation followed by RT-qPCR analyses revealed that *SUNO1* is a poly A⁺ RNA that is present in both the nucleus and cytoplasm (*Figure 2—figure supplement 2B–C*). Single-molecule (sm)-RNA-FISH revealed that *SUNO1* was preferentially enriched as 2–3 well-separated puncta in the nucleus (*Figure 2C*). The nuclear puncta signal, detected by the *SUNO1* smRNA-FISH probe set was absent in *SUNO1* knock-out (KO) cells, confirming the specificity of *SUNO1* localization (*Figure 2C*; *Figure 2—figure supplement 3A* for sm-FISH probe position and also the deleted region in the

CRISPR KO cells). Northern blot with a *SUNO1*-unique probe in BT-20 and HCT116 cells hybridized to discrete bands of >2 kb and >5 kb in length (*Figure 2—figure supplement 3B–C*). These bands were absent in *SUNO1* HCT116 KO cells, implying that *SUNO1* primarily codes for two isoforms. Publicly available RNA-seq data from multiple cell lines (BT-20 [*Ghandi et al., 2019*; *Varley et al., 2014*], HCT116 and MCF7 [*Andrysik et al., 2017*]) revealed >2 kb transcript to be the predominant isoform of *SUNO1,* with the higher molecular weight isoform present in lower levels, further confirming our Northern blot data (*Figure 2—figure supplement 3A & D*). Furthermore, GRO-seq (*Andrysik et al., 2017*), CAGE as well as poly A$^+$ seq data sets confirmed defined transcription start site (located within the CpG island) and the 3'end of *SUNO1* (*Figure 2—figure supplement 3D*). Estimation of protein-coding potential using PhyloCSF revealed that similar to the well-characterized *MALAT1* lncRNA, *SUNO1* did not show any protein-coding potential (*Figure 2—figure supplement 3E*). Finally, RNA stability assays revealed *SUNO1* to be a relatively stable poly A$^+$ RNA with a half-life of >2.6 hr (*Figure 2—figure supplement 3F*). Altogether, our results indicate that *SUNO1* is a G1/S to S-phase-induced low copy but relatively stable poly A$^+$ lncRNA and is preferentially enriched in the nucleus as 2–3 puncta.

## Depletion of *SUNO1* results in defective cell-cycle progression and hypersensitivity to DNA damage

We next determined whether *SUNO1* was required for normal cell-cycle progression. We successfully depleted *SUNO1* using multiple independent siRNAs targeting different regions of *SUNO1* (si*SUNO1*) in U2OS or HCT116 cells (*Figure 3A*, *Figure 3—figure supplement 1A* and also see *Figure 2—figure supplement 3A* for siRNA positions). *SUNO1*-specific siRNA-treated cells showed significant downregulation of both the nuclear and the cytoplasmic pool of *SUNO1* (*Figure 3—figure supplement 1B*). Propidium Iodide (PI)- as well as BrdU-PI-flow cytometry analyses revealed that *SUNO1*-depleted U2OS and HCT116 cells showed reduced number of cells in S-phase and a concomitant increase in G1 population, suggesting a defect in efficient progression into S-phase (*Figure 3a-b* and *Figure 3—figure supplement 1C a-b*). Furthermore, reduced number of cells in S-phase upon *SUNO1* depletion was confirmed by BrdU incorporation followed by immunostaining in control and *SUNO1*-depleted cells (*Figure 3—figure supplement 1D*). *SUNO1*-depleted cells also showed reduced cell proliferation compared to control cells, indicating that defects in cell-cycle progression upon *SUNO1* depletion contribute to defects in cell proliferation (*Figure 3C*). Finally, independent clones of *SUNO1* KO cells (both in HCT116 and U2OS) generated *via* CRISPR/Cas9-mediated genome-editing also displayed G1 or G1/S arrest and reduced cell proliferation (*Figure 3—figure supplement 1E–H*) similar to *SUNO1* knockdown cells, further supporting the involvement of *SUNO1* in S-phase entry.

S-phase of the cell cycle is an intrinsically challenging phase for cells, given that any defect during the initial stages of DNA replication could give rise to DNA damage that could induce G1 or G1/S arrest (*Macheret and Halazonetis, 2015*). In order to determine if the accumulation at G1 or G1/S observed upon *SUNO1* depletion is a result of enhanced DNA damage, we determined whether *SUNO1*-depleted cells were more prone to DNA damage. We found that *SUNO1*-depleted asynchronous cells showed significant increase in the levels of DNA damage as observed by DNA comet assays (*Figure 3—figure supplement 2Aa-b*). *SUNO1*-depleted cells also showed increased number of RPA32- (+ve cells, control = 7.2%; si*SUNO1*-a = 65.8%; si*SUNO1*-b 36.5%; n ≥ 75) and 53BP1- (+ve cells, control = 26.6%; si*SUNO1*-a = 53.5%; si*SUNO1*-b 64.5%; n ≥ 70) decorated nuclear foci, indicative of DNA damage (*Figure 3—figure supplement 2B*). Finally, *SUNO1*-depleted cells also showed increased levels of p53 as well as phospho-Chk2, consistent with increased DNA damage (*Figure 3—figure supplement 2C*). To specify whether the induction of p53 upon *SUNO1* depletion contributes to G1 or G1/S arrest, we determined the extent of G1 or G1/S arrest in *SUNO1*-depleted $p53^{+/+}$ or $p53^{-/-}$ HCT116 cells. PI-flow cytometry analyses revealed that unlike *p53* wild-type cells, *SUNO1*-depleted $p53^{-/-}$ HCT116 cells failed to arrest in G1 (*Figure 3—figure supplement 2D*) but showed increase in G2/M population. This result indicates that the G1 or G1/S arrest observed in *SUNO1*-depleted cells requires functional p53, implying *SUNO1*-depleted cells elicit intra-G1 or G1/S checkpoint.

The increased DNA damage observed in *SUNO1*-depleted cells prompted us to investigate whether *SUNO1* levels were sensitive to DNA damage. Cells treated with drugs that induced double-strand breaks such as doxorubicin (DNA intercalator and topoisomerase II inhibitor), or

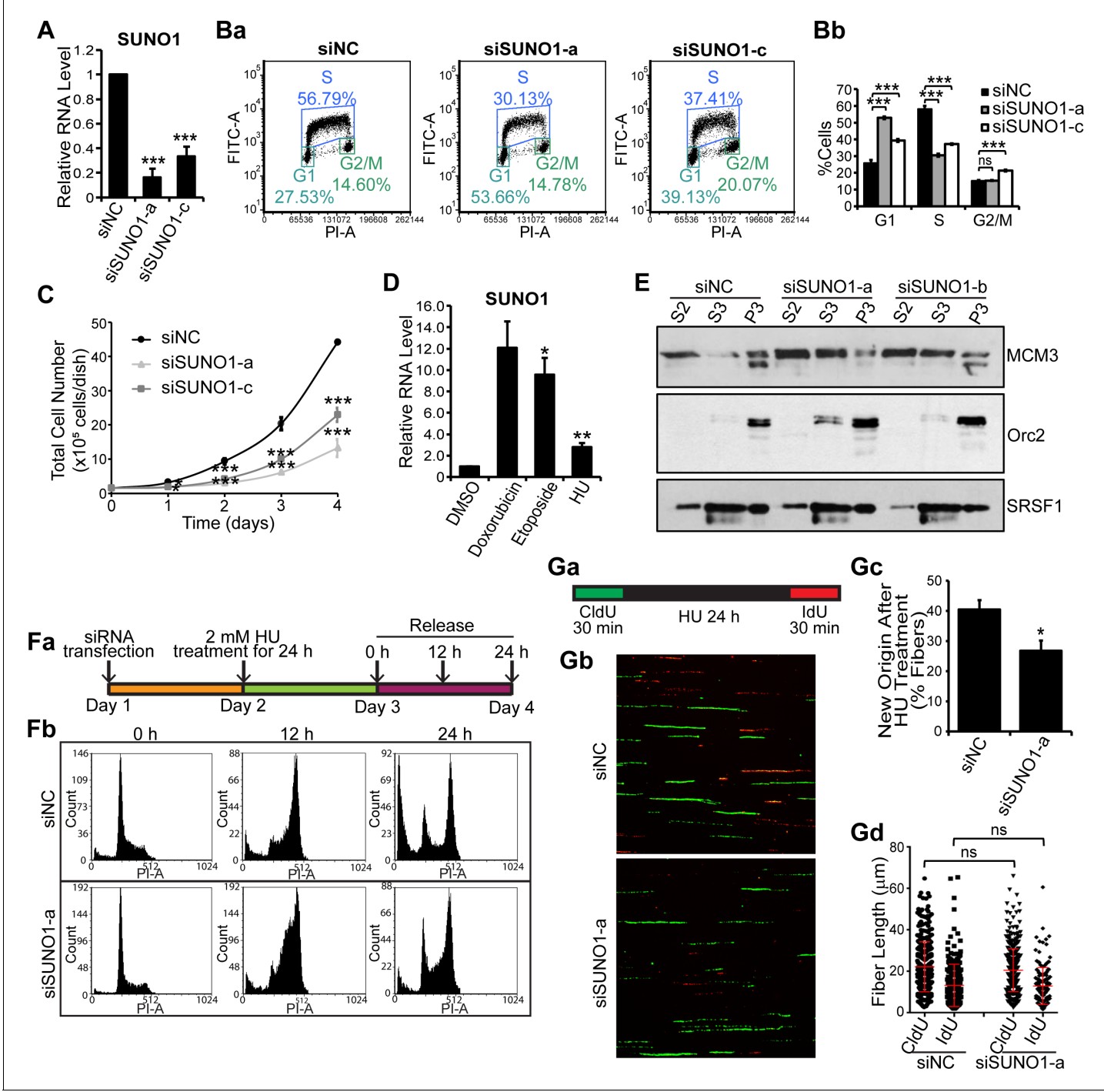

**Figure 3.** *SUNO1* depletion results in cell-cycle arrest and defects in S-phase entry. (**A**) RT-qPCR to quantify *SUNO1* levels in control (siNC) and *SUNO1*-specific siRNA (a and c)-treated HCT116 cells. Data are presented as Mean ± SD, n = 3. Unpaired two-tail t-tests are performed. *p<0.05, **p<0.01, ***p<0.001. (**B**) BrdU-PI-flow cytometry analyses of control (siNC) and *SUNO1*-specific siRNA (a and c)-treated HCT116 cells. Dot graphs from one of the replicates are shown (Ba). Population of G1, S and G2/M cells are quantified (Bb). Data are presented as Mean ± SD, n = 3. Unpaired two-tail t-tests are performed. ns, not significant; *p<0.05; **p<0.01; ***p<0.001. (**C**) Growth curve assay of control (siNC) and *SUNO1*-specific siRNA (a and c)-treated HCT116 cells. Data are presented as Mean ± SD, n = 3. Unpaired two-tail t-tests are performed. *p<0.05, **p<0.01, ***p<0.001. (**D**) RT-qPCR to quantify *SUNO1* levels in U2OS cells that are incubated with DMSO (control) and drugs (Doxorubicin [0.5 µM for 24 hr], Etoposide [20 µM for 24 hr] and Hydroxyurea [HU; 2 mM for 24 hr]), all of which induce double-strand DNA breaks. Data are presented as Mean ± SD, n = 3. Unpaired two-tail t-tests are performed. *p<0.05, **p<0.01, ***p<0.001. (**E**) Cellular fractionation to determine the chromatin loading of MCM3 and ORC2 in control (siNC) and *SUNO1*-depleted U2OS cells. S2 = cytoplasmic fraction; S3 = soluble nuclear fraction; P3 = insoluble chromatin fraction. SRSF1 is used as control. Refer

*Figure 3 continued on next page*

*Figure 3 continued*

to *Figure 3—source data 1*. (**Fa**) Flow chart showing the experimental plan. (**Fb**) PI-flow cytometry analyses to assess cell-cycle progression in U2OS cells transfected with siNC or si*SUNO1*-a, followed by 24 hr of 2 mM HU treatment, and released in fresh medium for 0, 12 and 24 hr. (**G**) Data from DNA fiber experiments in control and *SUNO1*-depleted U2OS cells. (**Ga**) DNA fiber experimental plan. DNA fiber experiments of U2OS cells treated with siNC or si*SUNO1*-a. U2OS cells are transfected with siNC or si*SUNO1*-a, pulse-labeled with CldU (green) for 30 min, followed by 24 hr of 2 mM HU treatment, and then released for 30 min in presence of IdU (red). DNA fiber spreads are prepared in biological triplicates. Representative images from one of the replicates are shown (**Gb**). The percentage of new origins (**Gc**) and the tract length of CldU and IdU fibers (**Gd**) are determined by counting 200 fibers per replicate. Data are presented as Mean ± SD, n = 3. Unpaired two-tail t-tests are performed. ns, not significant; *p<0.05; **p<0.01; ***p<0.001. Refer to *Figure 3—source data 2*.

The online version of this article includes the following source data and figure supplement(s) for figure 3:

**Source data 1.** Uncropped images of the Western Blot in *Figure 3E*, *Figure 3—figure supplement 2C*, and *Figure 3—figure supplement 3B*.
**Source data 2.** Quantification of the fiber assay in *Figure 3G*, *Figure 3—figure supplement 2C*, and *Figure 3—figure supplement 3B*.
**Figure supplement 1.** Depletion of *SUNO1* results in cell-cycle arrest and DNA damage.
**Figure supplement 2.** *SUNO*-1depleted cells show DNA damage.
**Figure supplement 3.** *SUNO1*-depleted cells are sensitive to drug-induced DNA damage.

Etoposide (topoisomerase II inhibitor), or hydroxyurea (HU; 2 mM for 24 hr for inducing replication fork collapse) (*Petermann et al., 2010*) showed pronounced induction of *SUNO1* (*Figure 3D* and *Figure 3—figure supplement 1F*). Several lncRNAs participate in the p53-mediated stress response, and their induction upon DNA damage is dependent on the integrity of the p53 pathway (*Huarte et al., 2010*; *Hung et al., 2011*). However, we observed a significant induction of *SUNO1* in both wild-type (WT; *p53 $^{+/+}$*) and *p53 $^{-/-}$* HCT116 cells upon DNA damage (data not shown). In addition, treatment of cells with Nutlin-3, a stabilizer of p53 did not induce *SUNO1*, further indicating that *SUNO1* activation was not mediated by p53 (*Figure 3—figure supplement 2E*).

We demonstrated that *SUNO1* was upregulated during S-phase and upon DNA damage, and loss of *SUNO1* led to cell-cycle arrest with increased DNA damage, resulting in cell-cycle checkpoint activation. One likely scenario is that *SUNO1* is required for entry into S-phase and perhaps for S-phase progression as well. Without *SUNO1*, the cells have difficulty entering S-phase and hence arrest at the G1/S boundary.

To gain molecular insights into why G1 accumulation is observed in *SUNO1*-depleted cells, we performed chromatin fractionation of pre-replicative complex proteins in control cells and in ones lacking *SUNO1*. It is known that defects in the pre-RC complex levels or their chromatin loading could compromise the origin assembly and/or firing. We observed reduced chromatin loading of the Mini Chromosome Maintenance 3 (MCM3), core component of the MCM helicase complex but not (Origin recognition complex 2) ORC 2, member of the ORC complex in *SUNO1*-depleted cells (*Figure 3E*), supporting the model that aberrant G1 or G1/S arrest observed upon *SUNO1* depletion could be partially due to defects in pre-RC assembly. This is consistent with our results that fewer origins are licensed in the absence of *SUNO1* leading to an accumulation in G1 phase.

To address why there was increased DNA damage in the absence of *SUNO1*, we addressed if *SUNO1* is involved in sensing and/or repairing DNA damage or the phenotype is a consequence of fewer licensed origins. To test this, we treated control and *SUNO1*-depleted or *SUNO1*-KO cells with HU for 24 hr, a condition that elicits strong replication stress by causing replication fork collapse (*Petermann et al., 2010*), and analyzed the recovery of cells post-HU-release (*Figure 3Fa* and *Figure 3—figure supplement 3A*). By PI-flow cytometry, we observed that control cells, post-HU-release, resumed DNA replication, with majority of them reaching G2/M phase by 12 hr post- HU-release (*Figure 3Fb* and *Figure 3—figure supplement 3A*). However, *SUNO1*-depleted and knock-out cells showed slow S-phase progression, as observed by the accumulation of a significant fraction of cells in S-phase 12 hr (hr) post-HU-release (*Figure 3Fb* and *Figure 3—figure supplement 3A*). Reduced S-phase progression upon *SUNO1* deletion could be due to the inability to repair DNA damage or due to defective fork progression. To test this, we performed a DNA fiber combing assay (*Figure 3G*). Control and *SUNO1*-depleted cells were first incubated with the thymidine analog 5-chloro-2'-deoxyuridine (CldU) for 30 min to label the replicating DNA strands. CldU was then washed off, and cells were treated with HU for 24 hr to induce replication fork collapse and double strand breaks. Then, cells were released into fresh medium containing another thymidine analog, 5-iodo-2'-deoxyuridine (IdU) for 30 min (*Figure 3Ga*). By this, the newly synthesized DNA strands will

be labeled with IdU. The DNA Fiber assay revealed that *SUNO1*-depleted cells showed reduced number of tracks that incorporated only IdU (red) post-HU treatment, implying a reduced number of dormant replication origins firing (*Figure 3Gb-c*). At the same time, both control and *SUNO1*-depleted cells showed comparable length in the CldU-labeled fibers (green), indicating a normal rate of replication fork progression (*Figure 3Gb and d*), suggesting that *SUNO1* is not required for S-phase progression, once the replication is initiated. Based on the results from the DNA fiber assay, we conclude that the defects in the S-phase progression observed in *SUNO1*-depleted cells post-HU-release are due to inefficient firing of dormant replication origins.

Interestingly, *SUNO1*-depleted cells failed to elicit some of the key DNA damage-induced checkpoint responses. For example, compared to control cells, *SUNO1*-depleted cells post-HU treatment (2 mM for 24 hr) showed reduced Chk1 phosphorylation at Ser345, BRCA1 phosphorylation at Ser1524, RPA32 phosphorylation and γH2AX induction, indicative of defective ATR-mediated checkpoint activation (*Figure 3—figure supplement 3B*). Furthermore, cell viability (MTT assay) and long-term cell survival assays (clonogenic assay) with and without DNA damage (Doxorubicin) revealed that *SUNO1* acted as a pro-survival gene. *SUNO1*-depleted cells showed reduced cell growth/survival under both normal and after DNA-damage conditions. (*Figure 3—figure supplement 3BCa-b*; data not shown), These data revealed that *SUNO1*-depleted cells are more sensitive to drug-induced DNA damage, implying that *SUNO1* is involved in DNA-damage response (DDR), and its loss causes defects in the cells' ability to recover from DNA damage. Our results demonstrate that *SUNO1* is required for entry into S-phase, and its depletion renders cells to become more sensitive to DNA damage, resulting in the inability to reinitiate DNA replication upon replicative stress.

## *SUNO1* regulates cell proliferation by promoting the expression of YAP1-target genes

In order to understand the underlying molecular mechanism by which nuclear-enriched *SUNO1* regulates cell proliferation, we analyzed the gene expression changes at the steady state levels in control and *SUNO1*-depleted cells. We isolated RNA from control and *SUNO1*-depleted HCT116 cells from early and late time points (36 and 72 hr after first round of siRNA treatment) and performed transcriptome-wide microarray analyses. Gene expression changes observed at the earlier time point would help to identify the primary targets of *SUNO1*. Cells collected 36 hr post siRNA treatment showed efficient depletion of *SUNO1* but did not show any observable cell-cycle defect phenotype, assessed by PI-flow cytometry analyses (data not shown). On the other hand, cells treated with SUNO1-specific siRNA for 72 hr showed pronounced cell-cycle arrest (*Figure 3B*). To identify primary targets of *SUNO1*, we looked for common target genes whose expression was altered in both early (36 hr) and late time (72 hr) points (*Supplementary file 6*). We observed 149 common genes that displayed reduced expression after 36 and 72 hr post *SUNO1* depletion (*Figure 4Aa* and *Supplementary file 6*). Further, Gene ontology (GO) analyses of genes that were downregulated even during early time point post *SUNO1* depletion (when there was no cell-cycle defect) revealed that they regulate cellular pathways, including Cellular Growth and Proliferation and Cell Death and Survival pathways (*Figure 4Ab*).

Since *SUNO1* appears to promote cell proliferation, we analyzed whether genes that are part of a particular cell growth controlling pathway were overrepresented in the list of 149 genes, the expression of which was altered upon *SUNO1* depletion. We observed that several known YAP1 (Yes-associated protein 1) target genes showed reduced expression in *SUNO1*-depleted cells (*Figure 4B*). YAP1 is a transcription co-activator that positively regulates TEAD- or FOS-mediated transcription of genes, thereby promoting cell proliferation (*Ehmer and Sage, 2016*). For example, *CCND1*, *CTGF*, *CYR61* and *AMOTL2* are the known targets of YAP1 (*Harvey et al., 2013*; *Zhao et al., 2008*), and we found that these genes were significantly downregulated in *SUNO1*-depleted cells (*Figure 4B*). In support of the gene expression data, *SUNO1*-depleted cells also showed reduced protein levels of YAP1 targets, including Cyclin D1, CTGF (*Figure 4C*). In addition, another potential YAP1 target, p15/PAF, a PCNA-associated factor that plays crucial roles in S-phase progression and DNA-damage repair (*Xie et al., 2014*; *Chang et al., 2013*; *De Biasio et al., 2015*; *Povlsen et al., 2012*; *Jung et al., 2013*), also showed reduced expression in *SUNO1*-depleted control and DNA-damaged cells (*Figure 4B* and *Figure 4—figure supplement 1A–B*). We consistently observed reduced mRNA and protein levels of YAP1 in *SUNO1*-depleted cells (*Figure 4B–C*). A recent study indicated that YAP1 positively autoregulates its own expression (*Vázquez-Marín et al., 2019*). In support of this

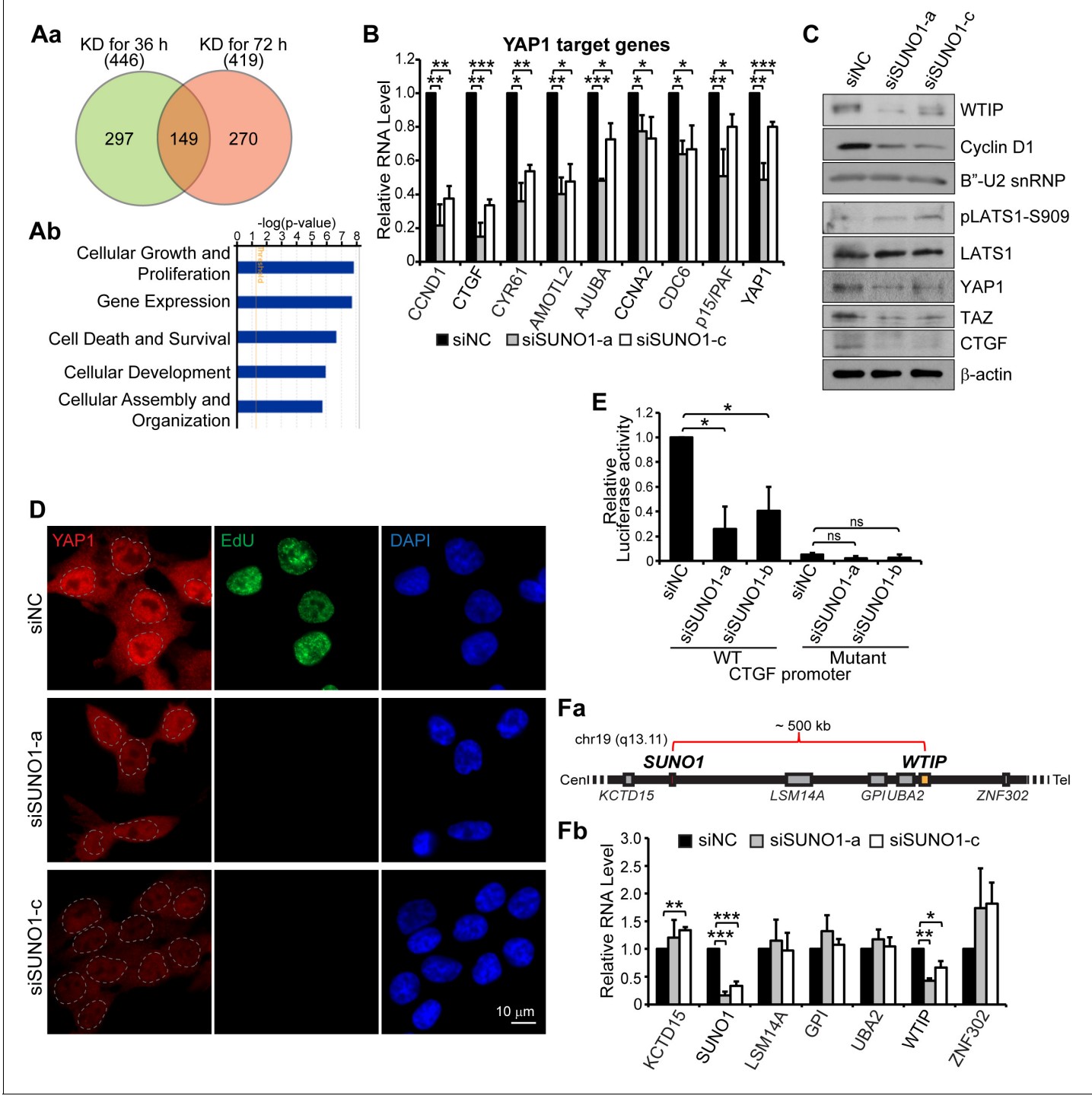

**Figure 4.** *SUNO1* promotes cell proliferation by regulating the expression of *WTIP*, a positive regulator of YAP1. (**Aa**) Venn diagram showing significantly downregulated genes in *SUNO1*-depleted wild-type (WT) HCT116 cells (36 and 76 hr post siRNA treatment). 149 common genes showed significant downregulation at both early (36 hr) and late (72 hr) time points post *SUNO1* depletion. (**Ab**) Gene ontology (GO) analysis of all of the genes downregulated after 36 hr post *SUNO1* knockdown. (**B**) RT-qPCR to show the levels of several YAP1 target gene mRNAs in control and *SUNO1*-depleted WT HCT116 cells. Data are presented as Mean ± SD, n = 3. Unpaired two-tail t-tests are performed. *p<0.05, **p<0.01, ***p<0.001. (**C**) Western blot to detect WTIP, Cyclin D1, YAP1, LATS1, pLATS1, TAZ, and CTGF in control and *SUNO1*-depleted WT HCT116 cell. B'-U2 snRNP and β-actin are used as loading control. Refer to *Figure 4—source data 1*. (**D**) Immunofluorescence staining to assess the cellular localization of YAP1 coupled with EdU incorporation assay. Cells in S-phase were labeled by EdU. Scale bar: 10 μm. (**E**) *CTGF* promoter luciferase assay. WT *CTGF* promoter (WT) or TEAD-binding sites mutated *CTGF* promoter (mutant) were cloned upstream of the luciferase reporter gene. WT or mutant reporters

*Figure 4 continued on next page*

*Figure 4 continued*
are transfected into control and *SUNO1*-depleted (si*SUNO1*-a or si*SUNO1*-b) U2OS cells, and the relative luciferase activity is quantified. Data are presented as Mean ± SD, n = 3. Unpaired two-tail t-tests are performed. ns, not significant; *p<0.05; **p<0.01; ***p<0.001. (**Fa**) Diagram showing relative genomic position of *SUNO1* and other genes near *SUNO1* locus. (**Fb**) RT-qPCR to show the relative mRNA levels from *SUNO1* and other genes that are located in the genomic proximity of *SUNO1* gene locus in control and *SUNO1*-depleted WT HCT116 cells. Data are presented as Mean ± SD, n = 3. Unpaired two-tail t-tests are performed. *p<0.05, **p<0.01, ***p<0.001.

The online version of this article includes the following source data and figure supplement(s) for figure 4:

**Source data 1.** Uncropped images of the Western Blot in *Figure 4C*, *Figure 4—figure supplement 1F*, and *Figure 4—figure supplement 2A*.
**Figure supplement 1.** *SUNO1*-depleted cells show defects in cell proliferation by downregulating the levels of YAP1 target genes.
**Figure supplement 2.** Stable overexpression of *WTIP* partially rescues the cell-cycle phenotype caused by *SUNO1* depletion.

observation, *YAP1* promoter contains several TEAD4 binding sites (data not shown), implying that *YAP1* expression could be regulated by TEAD/YAP1 axes. Finally, *SUNO1*-depleted cells also showed reduced levels of TAZ, the YAP1 paralog, which also promotes cell proliferation by co-activating the TEAD-mediated transcription (*Figure 4C*).

During cell cycle, active YAP1 protein is imported into the nucleus, where it positively regulates the TEAD- and FOS-mediated transcription of genes controlling cell proliferation (*Meng et al., 2016*; *Kim et al., 2019*). We therefore tested whether *SUNO1*-depleted cells alter the nuclear and cytoplasmic levels of YAP1 by immunostaining. Control cells showed nuclear as well as cytoplasmic distribution of YAP1 (*Figure 4D*). However, we observed decrease in the levels of YAP1, including the nuclear pool upon *SUNO1* depletion, implying that *SUNO1*-depleted cells reduced active pool of YAP1 (*Figure 4D*). It is established that phosphorylated LATS1 (at serine-909) kinase by phosphorylating YAP1, inhibits its nuclear import, ultimately resulting in the YAP1 degradation (*Meng et al., 2016*). We therefore quantified the pLATS1 levels in control and *SUNO1*-depleted cells. We observed increased levels of pLATS1 in *SUNO1*-depleted cells (*Figure 4C*). This result suggests that increased pLATS1 could also contribute to the reduced levels of YAP1 in *SUNO1*-depleted cells.

Next, to test whether the reduced expression of YAP1 target genes observed in *SUNO1*-depleted cells is due to G1 arrest, we examined the expression of several cell-cycle genes whose expression is controlled by other cell proliferation-promoting and cell-cycle-regulated TFs, such as E2Fs in control and *SUNO1*-depleted cells. We observed no significant changes in the levels of E2F target mRNA (*CDT1*, *E2F3* and *MCM6*) in *SUNO1*-depleted cells (*Figure 4—figure supplement 1C*). In addition, we also observed downregulation of YAP1 and its target mRNA like *CCND1* even in *SUNO1*-depleted HCT116 *p53*[-/-] cells (*Figure 4—figure supplement 1D*), where in the absence of p53, *SUNO1* depletion did not induce G1 arrest (*Figure 3—figure supplement 2D*), further supporting that the downregulation of YAP1 targets in *SUNO1*-depleted cells is not a consequence of G1 arrest.

Finally, to test the status of YAP1/TEAD-mediated transcription activity in presence or absence of *SUNO1*, we employed a reporter system where the *CTGF* promoter was cloned upstream of a luciferase reporter. In addition to the reporter with the wild-type *CTGF* promoter, a mutant reporter with TEAD-binding sites mutated in the *CTGF* promoter was used as negative control (*Zhao et al., 2008*). We transfected wild-type and mutant reporters into control and *SUNO1*-depleted U2OS cells and quantified the reporter activities by luciferase assay. The knock-down of *SUNO1* resulted in the significant decrease of luciferase activity driven by the *CTGF* wild-type promoter (*Figure 4E*). Notably, mutation of the TEAD sites itself caused a strong decrease of transactivation, which was not significantly further decreased by *SUNO1* knock-down (*Figure 4E*). Altogether, our data support the model that *SUNO1* promotes TEAD-mediated transcription *via* modulating YAP1 activity.

## SUNO1 promotes YAP1-mediated transcription of cell-cycle genes by regulating *WTIP* expression

*SUNO1* lncRNA is a low abundant transcript (based on RNA-seq analyses and smRNA-FISH) and is preferentially enriched as 2–3 nuclear puncta. We hypothesized that like several other low abundant lncRNAs, *SUNO1* could function in cis, *via* regulating the expression of protein-coding genes located at its genomic proximity (*Wang and Chang, 2011*). To test this, we analyzed the microarray data from control and *SUNO1*-depleted cells and determined potential changes in the expression of genes that were located near *SUNO1* genomic locus (~1 Mb window) (*Figure 4Fa*). Out of the six

protein-coding genes that are located near *SUNO1* locus, we observed consistent reduced expression of only the *WTIP (Wilms tumor 1-interacting protein)*, a gene located ~500 kb downstream of the *SUNO1* locus, in *SUNO1*-depleted cells (*Figure 4Fb*). Immunoblot analyses confirmed reduced WTIP protein in *SUNO1*-depleted cells (*Figure 4C*). Reduced levels of *WTIP* were also observed in *SUNO1*-depleted HCT116 *p53$^{-/-}$* cells, indicating that the change in *WTIP* levels was not a consequence of cell-cycle arrest at G1 (*Figure 4—figure supplement 1D*).

WTIP is a member of the mammalian Ajuba LIM family proteins, along with Ajuba and LIMD1 (*Das Thakur et al., 2010*). In *Drosophila*, *Ajuba* promotes cell proliferation by positively regulating YAP1 activity (*Das Thakur et al., 2010*). Ajuba LIM family proteins are adaptor proteins, which communicate cell adhesive events with nuclear responses to antagonize the LATS1-medited inhibitory phosphorylation of YAP1, thereby negatively regulating the Hippo signaling pathway (*Harvey et al., 2013*; *Das Thakur et al., 2010*). Ajuba LIM family proteins stabilizes YAP1 by negatively regulating the interaction between pLATS1 and YAP1 (*Harvey et al., 2013*). Given this crucial role of WTIP in regulating YAP1/Hippo signaling, we hypothesized that *WTIP* could be an important *cis* target of *SUNO1*, mediating *SUNO1*'s positive impact on cell proliferation. In support of this, we observed that *WTIP* expression was also regulated during cell cycle, with highest levels of WTIP mRNA and protein observed during G1/S and S-phases, a time window that coincided with the elevated levels of *SUNO1* (*Figure 4—figure supplement 1E–F*). Furthermore, depletion of *WTIP* resulted in downregulation of *YAP1*, and YAP1 target mRNAs, such as *CCND1* and *CTGF* (*Figure 4—figure supplement 1G*). Finally, both *WTIP*- and *SUNO1*-depleted cells showed similar cell-cycle phenotypes (G1 or G1/S arrest and reduced S-phase) (*Figure 4—figure supplement 1Ha-b*), implying potential epistatic regulation.

Finally, we have attempted to rescue the defects in cell cycle as well as cellular levels of YAP1 observed in *SUNO1*-depleted cells by stably overexpressing WTIP. To achieve this, we stably expressed a doxycycline (Dox)-inducible version of *EGFP-WTIP* cDNA (*Ibar et al., 2018*) in HCT116 cells. Upon treating the cells with Dox, we achieved stable overexpression of WTIP in control and *SUNO1*-depleted cells (*Figure 4—figure supplement 2A*). BrdU-PI-flow cytometry analyses revealed that overexpression of WTIP in cells depleted of *SUNO1* partially rescued the cell-cycle defects. We observed a significant reduction in the G1 population (with a concomitant increase in S population) in *SUNO1*-depleted cells overexpressing WTIP (*Figure 4—figure supplement 2Ba-b*). However, the *SUNO1*-depleted cells, overexpressing WTIP continued to show p53 induction, and the p53 levels were comparable to *SUNO1*-depleted cells with no *WTIP* overexpression (*Figure 4—figure supplement 2A*). The absence of a complete rescue of cell-cycle defects in *WTIP*-overexpressed cells could be attributed to p53-mediated checkpoint activation. These results suggest that the DNA-damage phenotype observed in *SUNO1*-depleted cells may not be entirely due to defects in the WTIP/YAP1 pathway.

Next, we tested whether overexpression of WTIP in *SUNO1*-depleted cells rescues the cellular pool of YAP1. Immunofluorescence imaging revealed that *SUNO1*-depleted cells overexpressing EGFP-WTIP showed significant increase in the cellular pool of YAP1 compared to *SUNO1*-alone depleted cells (*Figure 4—figure supplement 2C*). These results indicate that *SUNO1* modulates YAP1 levels by regulating the expression of *WTIP*.

## *SUNO1* promotes *WTIP* transcription via regulating DDX5-RNA polymerase II interaction on the chromatin

We proposed that physical association between *SUNO1* and *WTIP* genes would facilitate the recruitment of the low-copy *SUNO1* lncRNA to the *WTIP* gene locus for its regulatory function. Chromosome confirmation capture (3C) analyses revealed potential physical interaction between *SUNO1* and *WTIP* gene locus in a *SUNO1* lncRNA-independent manner (*Figure 5A*). The 3C data was further supported by the publicly available Hi-C data set in HCT116 showing that both *SUNO1* and WTIP genes are located within a single TAD (*Rao et al., 2017*; *Figure 5—figure supplement 1*). On the other hand, the negative control gene locus, located next to the *SUNO1* locus, but were part of a different TAD did not interact with the *SUNO1* (*Figure 5A* and *Figure 5—figure supplement 1*).

LncRNAs regulate the expression of genes by facilitating the recruitment or stabilization of TFs, co-factors, chromatin regulators or RNA-binding proteins to chromatin or RNA (*Sun et al., 2018a*; *Chen and Carmichael, 2010*). In order to determine the molecular mechanism utilized by *SUNO1* to promote *WTIP* transcription, we searched for *SUNO1*-interacting proteins that could regulate *WTIP*

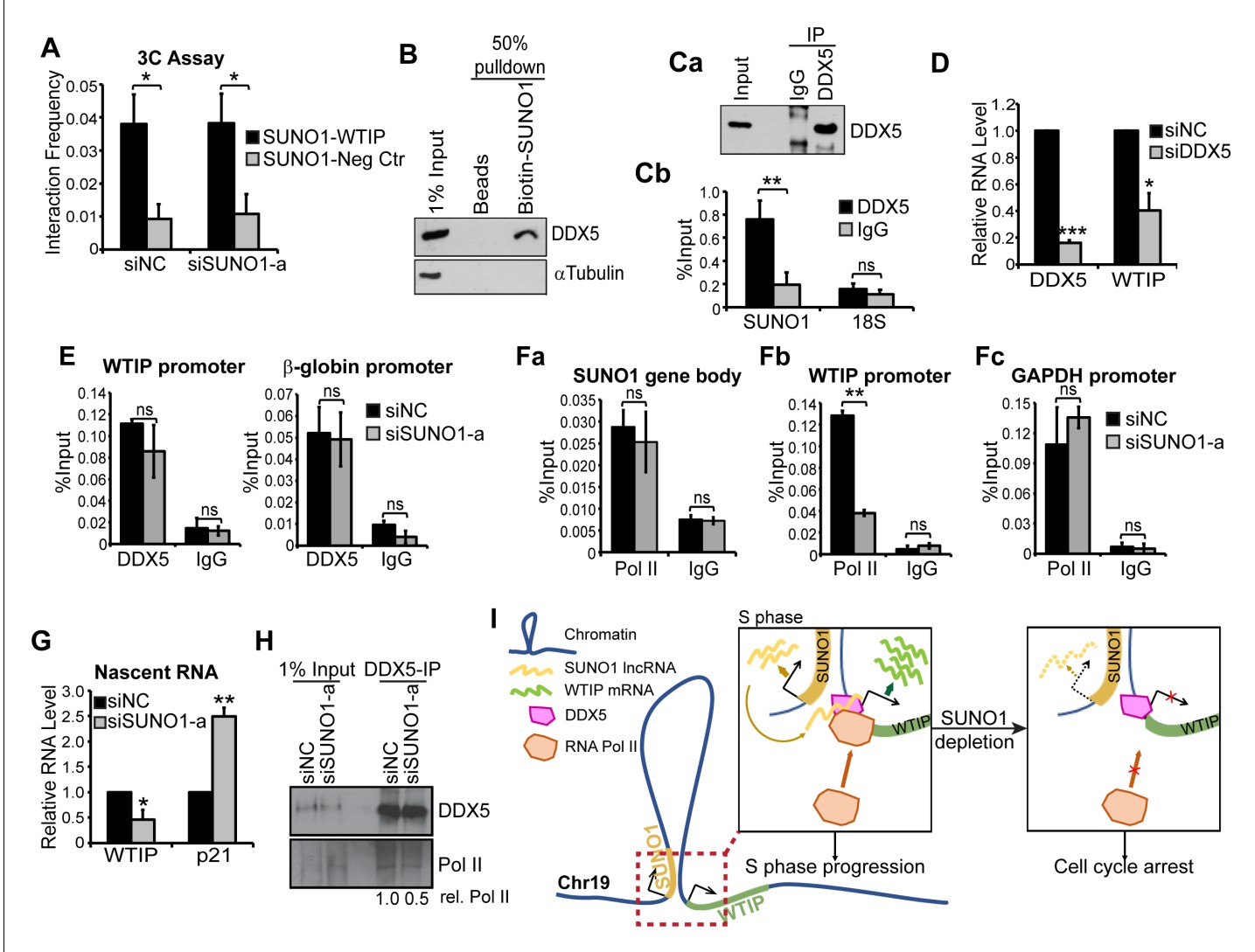

**Figure 5.** *SUNO1* promotes *WTIP* transcription by stabilizing the interaction between DDX5 and RNA polymerase II on chromatin. (**A**) 3C analyses to quantify the physical interaction frequency between *SUNO1* and *WTIP* genes in presence or absence of *SUNO1* RNA in WT HCT116 cells. Data are presented as Mean ± SD, n = 2. Unpaired one-tail t-tests are performed. *p<0.05, **p<0.01, ***p<0.001. (**B**) Western blot analysis to detect DDX5 and αTubulin in biotinylated RNA pulldown of *SUNO1* in WT HCT116 cells. αTubulin serves as a negative control. (**C**) DDX5-RIP in WT HCT116 cells followed by (**Ca**) western blot to detect DDX5 and (**Cb**) RT-qPCR to quantify the levels of *SUNO1* and 18S rRNA. 18S rRNA serves as a negative control for the binding of DDX5 to non-specific RNAs. Data are presented as Mean ± SD, n = 3. Unpaired two-tail t-tests are performed. ns, not significant; *p<0.05; **p<0.01; ***p<0.001. (**D**) RT-qPCR to quantify relative levels of DDX5 and *WTIP* mRNAs in control and DDX5-depleted WT HCT116 cells. Data are presented as Mean ± SD, n = 3. Unpaired two-tail t-tests are performed. *p<0.05, **p<0.01, ***p<0.001. (**E**) DDX5 ChIP-qPCR to quantify DDX5 association at the *WTIP* and *β-globin* promoter in control and *SUNO1*-depleted cells. IgG ChIP-qPCR on the same target genes serves as negative control. Data are presented as Mean ± SD, n = 3. Unpaired two-tail t-tests are performed. ns, not significant. (**F**) RNA pol II ChIP-qPCR to quantify RNA pol II association at the *SUNO1* gene body (**Fa**), *WTIP* promoter (**Fb**), and *GAPDH* promoter (**Fc**) in control and *SUNO1*-depleted cells. IgG ChIP-qPCR on the same target genes serves as negative control. Data are presented as Mean ± SD, n = 3. Unpaired two-tail t-tests are performed. ns, not significant; *p<0.05; **p<0.01; ***p<0.001. (**G**) Click-iT nascent RNA capture assays followed by RT-qPCR to quantify relative pre-mRNA levels of *WTIP* and *p21* in control *versus SUNO1*-depleted WT HCT116 cells. Note: increased levels of *p21* nascent RNA (a direct target of p53) in *SUNO1*-depleted cells confirm DNA-damage-induced p53-mediated check-point activation upon *SUNO*1 depletion. Data are presented as Mean ± SD, n = 3. Unpaired two-tail t-tests are performed. *p<0.05, **p<0.01, ***p<0.001. (**H**) DDX5-IP on chromatin followed by DDX5 and RNA pol II immunoblot assays to detect the relative levels of RNA pol II that are associated with DDX5 on chromatin in control and *SUNO1*-depleted WT HCT116 cells. (**I**) Model depicting the mode of action of *SUNO1* in regulating the transcription of *WTIP*. During S-phase, enhanced levels of *SUNO1* lncRNA promotes *WTIP* transcription by stabilizing the chromatin interactions between DDX5 and RNA pol II on promoters of genes such as *WTIP*. In the absence of *SUNO1*, *WTIP* transcription is compromised due to defects in the loading of RNA pol II.

The online version of this article includes the following figure supplement(s) for figure 5:

*Figure 5 continued on next page*

Figure 5 continued

**Figure supplement 1.** *SUNO1* and *WTIP* locate in a single TAD. Hi-C data covering *SUNO1*, *WTIP*, and the negative control genomic loci in HCT116 cells (*Rao et al., 2017*; GSE104334) is visualized by 3D Genome Browser (http://promoter.bx.psu.edu/hi-c/view.php).
**Figure supplement 2.** *SUNO1* interacts with DDX5.

transcription. For this, in vitro transcribed biotinylated- *SUNO1* RNA (2.1 Kb isoform) was incubated with cell lysate, then *SUNO1*-interacting proteins were pulled down by streptavidin affinity purification followed by mass spectrometry analysis. Biotin-labeled YFP RNA was used as negative control. We identified several proteins that were enriched in the *SUNO1* RNA pull down (*Supplementary file 7*). We focused on the interaction between *SUNO1* and one of its interactors, DDX5 (also known as p68), because of its known function as a transcription co-activator of cell-cycle genes (*Fuller-Pace, 2013*; *Figure 5—figure supplement 2A–B*). The interaction between DDX5 protein and *SUNO1* lncRNA was confirmed by western blot analysis (*Figure 5B*) as well as RNA-Immunoprecipitation (RIP) using antibody against DDX5 followed by RT-qPCR to detect *SUNO1* (*Figure 5Ca-b*). DDX5 is a DEAD box RNA helicase, and also acts as a transcriptional co-factor to modulate the activity of several cell proliferation-promoting TFs (*Fuller-Pace, 2013*). For example, DDX5 has been reported to promote E2F1-, p53-, Androgen receptor- and β-catenin-mediated transcription of genes controlling cell-cycle progression and DDR (*Nicol et al., 2013*; *Clark et al., 2013*; *Wagner et al., 2012*; *Bates et al., 2005*; *Mazurek et al., 2012*). In addition, studies have reported the involvement of ncRNAs in regulating the co-activator activity of DDX5 (*Caretti et al., 2006*). Based on this, we hypothesized that *SUNO1* may facilitate the DDX5-mediated transcription of *WTIP* during the cell cycle. Cells depleted of DDX5 showed reduced levels of *WTIP* mRNA, indicating that DDX5 positively regulates *WTIP* expression (*Figure 5D*). DDX5 ChIP-qPCR in control cells revealed the association of DDX5 to the *WTIP* and *β-globin* (positive control) promoters (*Figure 5E*). However, DDX5 continued to associate with both *WTIP* and *β-globin* promoters even in *SUNO1*-depleted cells, implying that *SUNO1* did not recruit/stabilize DDX5 to *WTIP* regulatory elements (*Figure 5E*). Recent studies have reported that DDX5 promotes the transcription of cell-cycle genes by recruiting or stabilizing RNA polymerase II (RNA pol II) (*Clark et al., 2013*; *Mazurek et al., 2012*; *Rossow and Janknecht, 2003*). We therefore quantified the RNA pol II association to *WTIP* promoter in the presence or absence of *SUNO1*. Initially, we determined the association of RNA pol II in the *SUNO1* gene body of cells treated with control siRNA as well as siRNA targeting the 3'end of *SUNO1*. ChIP-qPCR assay revealed that RNA pol II showed comparable levels of binding to the *SUNO1* gene body in control and *SUNO1* siRNA-treated cells (*Figure 5Fa*). These results imply that siRNA targeting the 3'end of the *SUNO1* gene (*Figure 2—figure supplement 3A* for si*SUNO1*a position) only degraded *SUNO1* lncRNA and did not affect the transcription from the *SUNO1* locus. In support of this, a recent study demonstrated that antisense oligonucleotides targeting the 3'end of the gene normally degrade only the transcript without impacting the transcription from the locus (*Lee and Mendell, 2020*). On the other hand, we observed significantly reduced association of RNA pol II to *WTIP* promoter in *SUNO1*-depleted cells (*Figure 5Fb*), compared to the control *GAPDH* promoter, which showed comparable levels of RNA pol II in control and *SUNO1*-depleted cells (*Figure 5Fc*). Nascent RNA capture followed by RT-qPCR revealed that *SUNO1*-depleted cells showed a significant reduction in the levels of nascent *WTIP* pre-mRNA (*Figure 5G*), further supporting the earlier result that *SUNO1* depletion reduced RNA pol II activity at *WTIP* locus. Increased levels of *p21* pre-mRNA observed in *SUNO1*-depleted cells, due to p53-mediated G1 checkpoint activation, was used as a positive control. We then examined whether *SUNO1* influenced the DDX5-mediated recruitment/stabilization of RNA pol II to gene promoters. Towards this, we tested the DDX5-RNA pol II interaction on chromatin in control versus *SUNO1*-depleted cells by DDX5-chromatin-IP in formaldehyde-crosslinked cell lysate followed by immunoblot assays. Control cells showed specific interaction between DDX5 and RNA pol II on chromatin (*Figure 5H*). However, *SUNO1*-depleted cells significantly compromised the interaction between DDX5 and RNA pol II on chromatin (*Figure 5H*). Based on this, we conclude that *SUNO1* lncRNA influences DDX5-mediated recruitment/stabilization of RNA pol II on the promoter in cis, thereby enhancing *WTIP* transcription (*Figure 5I*).

### *SUNO1* promotes tumorigenicity in colon cancer cells

Our results indicate a pro-proliferative function of *SUNO1*. Since we demonstrated that *SUNO1* facilitates the well-established oncogene YAP1-mediated transcription of genes promoting cell proliferation in colon carcinoma cells (HCT116), we wondered whether *SUNO1* contributes to tumor progression. Patient survival analyses using the colon adenocarcinoma samples from the TCGA data set revealed that patients with higher *SUNO1* levels displayed significantly shorter survival compared to patients with lower *SUNO1* expression, indicating that a high *SUNO1* level is associated with poor prognosis in colon adenocarcinoma (*Figure 6A*). Next, we tested whether the ~149 genes that showed reduced expression in *SUNO1*-depleted cells also exhibited synchronous change in expression patterns in the TCGA colon cancer patient cohort. Interestingly, a major fraction of these genes (71%), including *WTIP*, showed positive correlation in expression with SUNO1 across colon cancer patients, implying that *SUNO1* potentially regulates the expression of these genes even in cancer tissue samples (*Figure 6B* and *Figure 6—figure supplement 1*). Finally, we also observed a positive correlation between the levels of *SUNO1* and a significant number of YAP1-target mRNAs in the same patients, supporting our data that *SUNO1* regulates YAP1-mediated transcriptional program (*Figure 6C*; *Zhao et al., 2008*; *Kapoor et al., 2014*; *Shao et al., 2014*; *Zanconato et al., 2015*).

Further, to test the involvement of *SUNO1* in tumor progression, we performed anchorage-independent growth assays in wild-type and *SUNO1*-KO HCT116 cells. In contrast to wild-type HCT116 cells, *SUNO1*-KO cells significantly lost their ability to form colonies in soft agar, revealing the requirement of *SUNO1* for the tumorigenicity of HCT116 cells under in vitro conditions (*Figure 6D*). We next performed tumor xenograft assay to examine the effect of *SUNO1* deletion on primary tumor growth in vivo. SUNO1-KO HCT116 and control HCT116 cells were injected subcutaneously into the flanks of immune compromised mice, and the tumor sizes were monitored for 25–30 days post-injection. The tumor growth in *SUNO1*-KO cells was significantly compromised compared to control HCT116 cells (*Figure 6E*). These data collectively support the model that *SUNO1* participates in tumorigenesis.

## Discussion

In this study, we performed a comprehensive analysis to understand human lncRNA expression during the cell cycle. We identified >2000 lncRNAs with periodic expression patterns peaking at a specific cell-cycle phase. To demonstrate that the cell-cycle phase-specific expressed lncRNAs regulate vital cellular processes, we characterized the function of *SUNO1*, an S-phase-enriched lncRNA in cell proliferation. We observed that *SUNO1* regulated the expression of *WTIP*, a member of AJUBA family of proteins that repress Hippo signaling pathway. Furthermore, we have provided evidence indicating that *SUNO1* promoted transcription by facilitating the co-activator, DDX5-mediated recruitment/stabilization of RNA pol II on chromatin.

DDX5 is an established RNA helicase involved in multiple processes of RNA metabolism, including pre-mRNA splicing, rRNA and miRNA processing (*Fuller-Pace, 2013*). In addition, it is becoming increasingly clear that DDX5 also acts as transcription co-activator or co-repressor in a context-dependent manner *via* interacting with specific TFs or RNA pol II (*Fuller-Pace, 2013*). For example, in response to DNA damage, it interacts with and co-activates p53 to mediate cell-cycle arrest (*Nicol et al., 2013*). However, during normal cell-cycle progression, DDX5 stimulates the recruitment/stabilization of RNA pol II to the promoters of E2F1-regulated DNA replication factor genes, thereby promoting cell proliferation (*Mazurek et al., 2012*). Several other studies have also demonstrated the involvement of DDX5 in regulating RNA pol II activity, though the exact mechanism is yet to be established (*Clark et al., 2013*; *Rossow and Janknecht, 2003*). Interestingly, DDX5 is known to interact with ncRNAs (*Caretti et al., 2006*; *Das et al., 2018*). DDX5 facilitates the transcriptional activity of MyoD by forming a complex with the ncRNA *SRA* in muscle cells (*Caretti et al., 2006*). In the present study, we demonstrated that early S-phase- upregulated *SUNO1*, by forming a complex with DDX5, promotes the association between DDX5 and RNA pol II on chromatin, thereby promoting transcription of genes such as *WTIP*. Reduced *WTIP* mRNA level in DDX5-depleted cells further support the role of DDX5 as a regulator of *WTIP* transcription. Future studies will address how *SUNO1* influences the DDX5-mediated recruitment of RNA pol II specifically at the *WTIP* or other gene promoters. It is possible that the *SUNO1*-DDX5 RNP complex at *WTIP* promoter may either confer specificity in recruiting RNA pol II to *WTIP* promoter, and/or stimulate the transcriptional co-

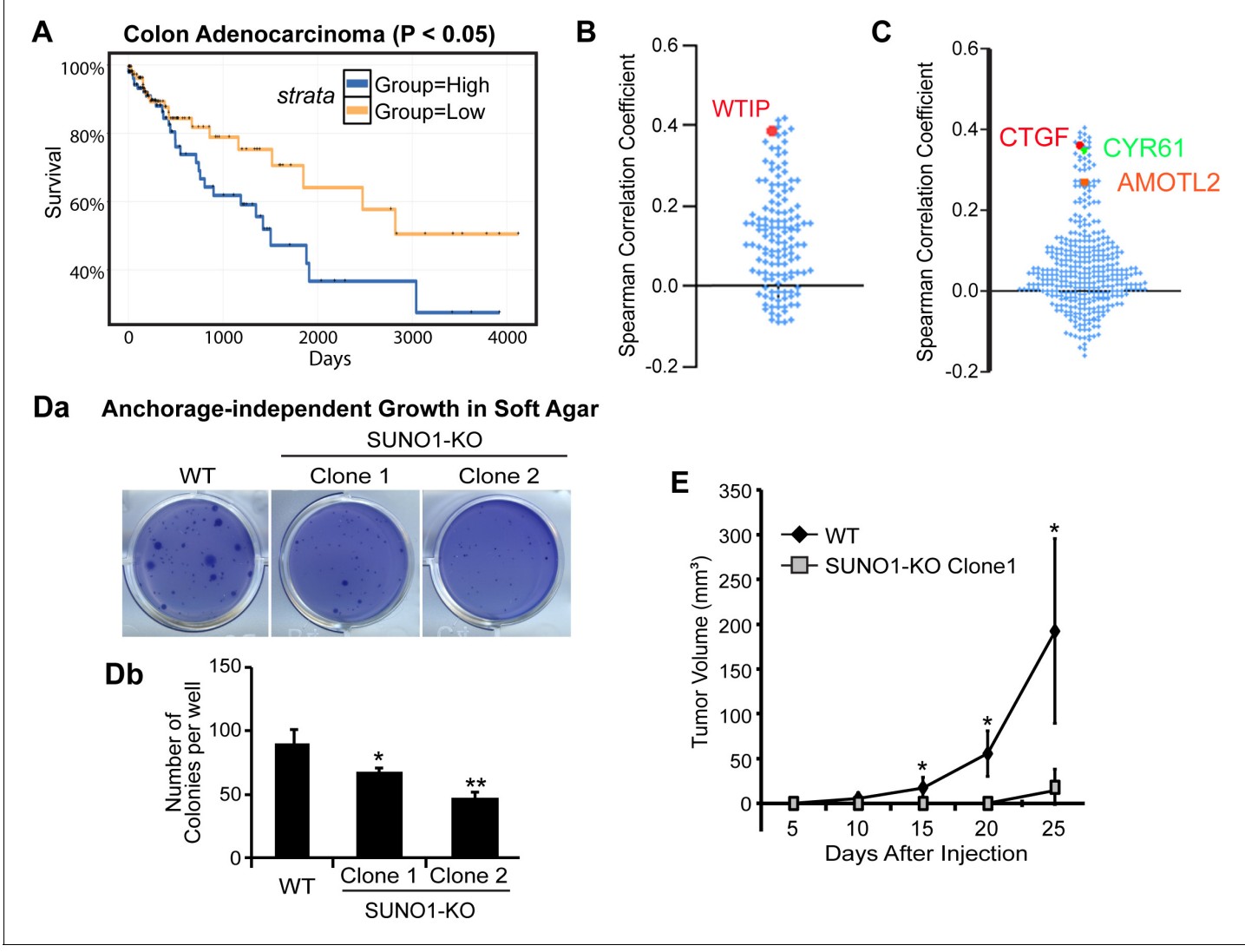

**Figure 6.** *SUNO1* contributes to tumorigenicity under in vitro and in vivo conditions. (**A**) Kaplan-Meier analyses to depict the survival rate in TCGA colon adenocarcinoma patients with high and low levels of *SUNO1*. Expression levels are separated into high and low levels across cancer samples based on median. (**B**) Spearman correlation of the expression levels of the 149 genes that are downregulated in *SUNO1*-depleted cells (*Figure 4A*; *Supplementary file 6*) with *SUNO1* in colon adenocarcinoma patient tumor samples. Each dot represents one of the downregulated genes upon *SUNO1* knockdown, and its Spearman correlation coefficient with *SUNO1* is plotted. All of the included positively correlated genes with *SUNO1* exhibited a p-value<0.01 at a 5% FDR. *WTIP* is highlighted in red. (**C**) Spearman correlation of the expression levels of YAP1/TAZ/TEAD target genes with *SUNO1* in colon adenocarcinoma patient tumor samples. Each dot represents one of the YAP1/TAZ/TEAD direct target genes, and its Spearman correlation coefficient with *SUNO1* is plotted. *CTGF, CYR61* and *AMOTL2* is highlighted. (**Da–b**) Long-term anchorage-independent colony formation assay in soft agar of wild-type and *SUNO1*-CRISPR KO HCT116 (Clone one and Clone 2) cells. (**E**) Tumor formation of wild-type control and *SUNO1*-CRISPR KO HCT116 (clone 1) cells in mouse xenograft experiments. Data are presented as Mean ± SD, n = 5. Paired two-tail t-tests are performed. *p<0.05, **p<0.01, ***p<0.001.

The online version of this article includes the following figure supplement(s) for figure 6:

**Figure supplement 1.** Distribution of spearman correlation values for various genes with respect to *SUNO1* across the colon adenocarcinoma cancer samples from the TCGA project.

activator activity of DDX5. Earlier studies, demonstrating the role of the ncRNA, *SRA* in promoting DDX5 activity support such a model (*Caretti et al., 2006*). In addition, a recent study showed that the *CONCR* lncRNA interacts with another helicase, DDX11, and regulates its enzymatic activity (*Marchese et al., 2016*). We therefore speculate that the mode of action of *SUNO1* may represent a wider spread mechanism in which lncRNAs interact with DEAD box family DNA/RNA helicases to modulate their location and activity.

The Hippo pathway controls organ size and tissue homeostasis in diverse species through regulating cell proliferation, apoptosis and stemness, whereas its deregulation contributes to tumor progression in a broad range of human carcinomas. Despite the fact that Hippo pathway activity is frequently deregulated in different human cancers, somatic or germline mutations in Hippo pathway genes are uncommon (*Harvey et al., 2013*; *Yu et al., 2015*). Here, by identifying the lncRNA *SUNO1* as a *cis* activator of *WTIP*, that positively regulates YAP1, we hypothesize that *SUNO1* acts as an oncogene *via* inhibiting the Hippo pathway. Our hypothesis is supported by the observation that elevated expression of *SUNO1* correlates with poor prognosis in colon adenocarcinoma, and further tumor assays revealed that *SUNO1* is required for the tumorigenicity of colon cell lines.

We observed that *SUNO1* was induced upon DNA damage. Furthermore, *SUNO1*-depleted cells showed slow S-phase progression post release from HU-mediated DNA damage, due to defects in replication origin re-activation. Also, *SUNO1*-depleted cells showed enhanced sensitivity to DNA damage. At present, the role of *SUNO1* in DDR is unclear. Interestingly, *SUNO1*-depleted cells failed to activate ATR-mediated DNA-damage checkpoint during HU treatment, as observed by reduced phosphorylation of several of ATR and CHK1 substrates. This could be due to the fact that *SUNO1*-depleted cells fail to enter S-phase as the ATR-mediated check point is active during S and G2 phase of the cell cycle (*Buisson et al., 2015*; *Zeman and Cimprich, 2014*). We propose that *SUNO1* contributes to DDR by modulating the expression of genes that regulate DDR. In support of this, *SUNO1*-depleted cells showed reduced expression of several genes (*GADD45B, CEBPA, UHRF1, P51/PAF*) that contribute to DDR (data not shown) (*Supplementary file 6*), though the mode of action is yet to be determined. Alternatively, the above-described phenotypes observed in *SUNO1*-depleted cells could be a consequence of aberrant replication stress. Dormant replication origins are activated following replication stress to ensure completion of DNA replication at stalled forks (*Zeman and Cimprich, 2014*). However, *SUNO1*-depleted cells showed reduced number of new origins firing post-HU treatment, and this could result in delayed S-phase progression. Future studies will test whether *SUNO1* actually plays a role in DDR or the aberrant DNA-damage phenotype observed in *SUNO1*-depleted cells is a consequence of error in origin licensing.

LncRNAs regulate cell proliferation and survival, by regulating the expression of cell-cycle-regulated protein-coding genes, such as cyclins or CDKs or CDK inhibitors (*Kitagawa et al., 2013*). Also, lncRNAs that are transcribed from the promoters of cell-cycle regulators have coordinated transcription of their respective protein-coding gene partners (*Hung et al., 2011*). A recent study using nascent DNA strand sequencing, identified >1000 s of lncRNAs to be induced during S-phase, and a significant number of these RNAs showed differential expression in pan-cancer samples (*Ali et al., 2018*). Further, loss-of-function studies revealed that several of the lncRNAs contribute to cancer progression, underpinning the important roles played by cell-cycle-regulated ncRNAs in cancer (*Ali et al., 2018*). Similarly, an independent study from the RIKEN group reported that depletion of a significant number of lncRNAs resulted in cell-cycle defects, further supporting the involvement of lncRNAs in cell-cycle progression (*Ramilowski et al., 2020*). These individual examples, though strengthened the argument about the importance of lncRNAs in cell cycle, failed to provide a genome-wide understanding of the crucial roles played by thousands of uncharacterized lncRNAs in cell proliferation. We have identified several hundreds of lncRNAs that displayed cell-cycle phase-specific expression. As a proof of principle, we have demonstrated the vital role for *SUNO1* in promoting YAP1-mediated expression of genes controlling cell proliferation. It is evident that similar to proteins, lncRNAs could constitute organized programs of biological activities that are required for efficient cell proliferation. Our study would be the first step in the continuum of research that is expected to lead to the functional characterization of a large number of cell-cycle-regulated lncRNAs.

# Materials and methods

**Key resources table**

| Reagent type (species) or resource | Designation | Source or reference | Identifiers | Additional information |
|---|---|---|---|---|
| Antibody | Anti-BrdU (Mose monoclonal) | Sigma-Aldrich | B9434 | IF (1:800) |
| Antibody | Anti-MCM3 (Rabbit polyclonal) | Stillman B. lab, CSHL | clone 738 | WB (1:1000) |
| Antibody | Anti-Orc2 (Rabbit polyclonal) | Stillman B. lab, CSHL | clone 205–6 | WB (1:500) |
| Antibody | Anti-SRSF1 (Mouse monoclonal) | Krainer A. lab, CSHL | clone 96 | WB (1:1000) |
| Antibody | Anti-p53 (Mouse monoclonal) | Santa Cruz | sc-126 | WB (1:500) |
| Antibody | Anti-B'-U2 snRNP (Mouse polyclonal) | Spector lab, CSHL | clone 4G3 | WB (1:250) |
| Antibody | Anti-Chk1 (Rabbit polyclonal) | Cell Signaling | #2345 | WB (1:500) |
| Antibody | Anti-pChk1-S345 (Rabbit polyclonal) | Cell Signaling | #2348 | WB (1:500) |
| Antibody | Anti-Chk2 (Rabbit polyclonal) | Cell Signaling | #2662 | WB (1:500) |
| Antibody | Anti-pChk2-T68 (Rabbit polyclonal) | Cell Signaling | #2661 | WB (1:500) |
| Antibody | Anti-pBRCA1-S1524 (Rabbit polyclonal) | Cell Signaling | #9009 | WB (1:400) |
| Antibody | Anti-RPA32 (Rat polyclonal) | Cell Signaling | #2208 | WB (1:700), IF (1:500) |
| Antibody | Anti-γH2AX (Rabbit monoclonal) | Cell Signaling | #9718 | WB (1:700) |
| Antibody | Anti-αTubulin (Mouse monoclonal) | Sigma-Aldrich | T5168 | WB (1:5000) |
| Antibody | Anti-WTIP (Mouse polyclonal) | Sigma-Aldrich | SAB1411722 | WB (1:200) |
| Antibody | Anti-Cyclin D1 (Rabbit polyclonal) | Cell Signaling | #2922 | WB (1:500) |
| Antibody | Anti-LATS1 (Mouse monoclonal) | Santa Cruz | sc-398560 | WB (1:100) |
| Antibody | Anti-pLATS1-S909 (Rabbit polyclonal) | Cell Signaling | #9157 | WB (1:1000) |
| Antibody | Anti-YAP1 (Mouse monoclonal) | Santa Cruz | sc-376830 | WB (1:100), IF (1:50) |
| Antibody | Anti-TAZ (Mouse monoclonal) | Santa Cruz | sc-518036 | WB (1:100) |
| Antibody | Anti-CTGF (Mouse monoclonal) | Santa Cruz | sc-365970 | WB (1:100) |
| Antibody | Anti-β-Actin (Mouse monoclonal) | Santa Cruz | sc-47778 | WB (1:300) |
| Antibody | Anti-p15/PAF (Rabbit polyclonal) | Santa Cruz | sc-9996 | WB (1:200) |
| Antibody | Anti-GFP (Mouse monoclonal) | Santa Cruz | sc-67280 | WB (1:100) |
| Antibody | Anti-DDX5 (Mouse monoclonal) | Millipore | clone204, #05–580 | WB (1:200) |

*Continued on next page*

*Continued*

| Reagent type (species) or resource | Designation | Source or reference | Identifiers | Additional information |
|---|---|---|---|---|
| Antibody | Anti-Pol II (Mouse monoclonal) | Millipore | clone CTD4H8, #05–623 | WB (1:1000), ChIP (5 μg/experiment) |
| Antibody | Anti-53BP1 (Rabbit polyclonal) | Cell Signaling | #4937 | IF (1:300) |
| Antibody | Anti-DDX5 (Rabbit polyclonal) | BETHYL | A300-523A | ChIP (5 μg/experiment) |
| Antibody | Anti-BrdU (CldU) (Rat monoclonal) | Bio-Rad | OBT0030G, Clone BU1/75 (ICR1) | DNA fiber assay (1:200) |
| Antibody | Anti-BrdU (IdU) (Mouse monoclonal) | BD | #347580, clone B44 | DNA fiber assay (1:200) |
| Transfected construct | pT3.5 Caggs-FLAG-hCas9 | This paper | | Construct to express Cas9 for making KO cell lines |
| Transfected construct | pCR4-TOPO-U6-gRNA | This paper | | Backbone of the construct to express gRNAs for making KO cell lines |
| Transfected construct | pcDNA-PB7 | This paper | | Construct to express Piggy Bac transposase for making KO cell lines |
| Transfected construct | pPBSB-CG-Luc-GFP-Puro | This paper | | Construct to express the puromycin resistent gene for making KO cell lines |
| Transfected construct (human) | pTRIPZ-EGFP:WTIP | Addgene *Ibar et al., 2018* | #66953 | Lentiviral vector for Tet-inducible EGFP:WTIP fusion protein expression |
| Commercial assay or kit | FITC BrdU Flow Kit (RUO) | BD Pharmingen | #559619 | |
| Commercial assay or kit | ChIP-IT High Sensitivity kit | Active Motif | #53040 | |
| Commercial assay or kit | CometAssay Kit | Trevigen | 4250–050 K | |
| Commercial assay or kit | Dual-Luciferase Reporter Assay System | Promega | E1910 | |
| Commercial assay or kit | Click-iT Nascent RNA Capture Kit | Invitrogen | C10365 | |
| Commercial assay or kit | FiberPrep (DNA Extraction Kit) | Genomic vision | EXTR-001 | |
| Cell line (*H. sapiens*) | HCT116 | ATCC | CCL-247 | |
| Cell line (*H. sapiens*) | BT20 | ATCC | HTB-19 | |
| Cell line (*H. sapiens*) | U2OS | ATCC | HTB-96 | |
| Cell line (*H. sapiens*) | HeLa | ATCC | CCL-2 | |
| Cell line (*H. sapiens*) | HCT116 p53 -/- | Vogelstein B. lab, Johns Hopkins Uni. | | |
| Chemical compound, drug | Thymidine | Sigma-Aldrich | T9250 | |
| Chemical compound, drug | Nocodazole | Sigma-Aldrich | M1404 | |
| Chemical compound, drug | Doxorubicin hydrochloride | Sigma-Aldrich | D1515 | |
| Chemical compound, drug | Etoposide | Sigma-Aldrich | E1383 | |

*Continued on next page*

*Continued*

| Reagent type (species) or resource | Designation | Source or reference | Identifiers | Additional information |
|---|---|---|---|---|
| Chemical compound, drug | Hydroxyurea | Sigma-Aldrich | H8627 | |
| Chemical compound, drug | Nutlin-3 | Sigma-Aldrich | N6287 | |
| Chemical compound, drug | Actinomycin D | Sigma-Aldrich | A9415 | |
| Chemical compound, drug | Doxycyline Hyclate | Sigma-Aldrich | D9891 | |
| Chemical compound, drug | BrdU | Sigma-Aldrich | B9285 | |
| Chemical compound, drug | EdU | Invitrogen | A10044 | |
| Chemical compound, drug | CldU | Sigma-Aldrich | C6891 | |
| Chemical compound, drug | IdU | MP Biomedicals | SKU02100357.2 | |
| Chemical compound, drug | Alexa Fluor 488 Azide | Invitrogen | A10266 | |
| Sequence-based reagent | SUNO1-5'gRNA | This paper | gRNA for SCRISPR KO | CCTAACCTAGATCTCCC |
| Sequence-based reagent | SUNO1-3'gRNA | This paper | gRNA for SCRISPR KO | AGGGTGGACAGGGATGC |
| Sequence-based reagent | SUNO1-F | This paper | qPCR primers | CACCAACAGACGTGAGTTCGA |
| Sequence-based reagent | SUNO1-R | This paper | qPCR primers | AGAACACTGCGAGGCTCACA |
| Sequence-based reagent | siNC | This paper | control siRNA | targeted sequence: UUCUCCGAACGUGUCACGU |
| Sequence-based reagent | siSUNO1-a | This paper | SUNO1-specific siRNA | targeted sequence: GCACGUGGUAAUACAUAAU |
| Sequence-based reagent | siSUNO1-b | This paper | SUNO1-specific siRNA | targeted sequence: GAGGAAUGCUGAUCUAGAA |
| Sequence-based reagent | siSUNO1-c | This paper | SUNO1-specific siRNA | targeted sequence: GGCGUGAUUUAGAUGGAAA |
| Transfected construct (Human) | siRNA to WTIP (SMARTpool) | Dharmacon | L-023639-02-0005 | |

## Cell lines

U2OS and HeLa cells were grown in DMEM medium. HCT116 WT and $p53^{-/-}$ cells were grown in McCoy's 5A medium. BT-20 cells were grown in EMEM medium. All media were supplemented with 10% fetal bovine serum (FBS) and penicillin/streptomycin. Cells were maintained in a 5% $CO_2$ incubator at 37˚C. Cell lines are obtained from commercial vendors such as ATCC. We confirm that the identity of all cell lines used in our study has been authenticated by STR profiling. All cell lines were checked for mycoplasma.

## Generation of *SUNO1* CRISPR KO cell lines

*SUNO1* CRISPR KO HCT116 and U2OS clones were made by transiently transfecting pT3.5 Caggs-FLAG-hCas9, gRNA expressing plasmids (in pCR4-TOPO-U6-gRNA), PiggyBac Transposase expressing plasmid (pcDNA-PB7) and pPBSB-CG-Luc-GFP-Puro. Selection was carried out with 2 µg/ml of puromycin followed by single clone selection. The KO clones were confirmed by PCR followed by DNA sequencing.

## Generation of stable cell lines

pTRIPZ-EGFP:WTIP was a gift from Kenneth Irvine (*Ibar et al., 2018*; Addgene plasmid #108231). HCT116 cells were incubated with the lentiviral particles for 2 days. Cells were then selected in medium containing 1 µg/ml puromycin for 3 days. EGFP-WTIP was induced by adding 0.05 µg/ml of Doxycycline (DOX) 24 hr prior to siRNA transfection.

## Cell synchronization

U2OS cells were synchronized to different cell-cycle stages as previously described (*Tripathi et al., 2013*). Briefly, cells were synchronized to mitosis by treatment with 50 ng/ml nocodazole for 12 hr. To collect cells in G1 phase, mitotic cells were shaken off and released in fresh medium for 3.5 hr. G1/S-boundary, S-phase and G2-phase samples were collected by double-thymidine block and release. G1/S samples were collected after the second block. Cells were then released in fresh medium for 4 hr to be collected as S-phase samples and 8 hr to be collected as G2-phase samples.

## RNA extraction and quantitative real-time PCR (RT-qPCR)

RNA was extracted using Trizol reagent (Invitrogen) as per manufacturer's instructions. Samples for RNA-seq were further cleaned up by RNeasy Mini Kit (QIAGEN). RNA was reverse transcribed into cDNA by Multiscribe Reverse Transcriptase and Random Hexamers (Applied Biosystems). One-step RT-PCR was performed as previously described (*Sun et al., 2018b*; *Caretti et al., 2006*).

## Bioinformatics and statistical analyses of RNA-seq data

The RNA-seq libraries were prepared with Illumina's 'TruSeq Stranded mRNAseq Sample Prep kit' (Illumina). Paired-end, polyA+ RNA-sequencing was performed on Illumina platform (Novaseq 6000, SP flowcell) at the Roy J. Carver Biotechnology Center at UIUC. The RNA-seq are deposited in GEO with accession number GSE143275. High quality of RNA-seq reads was confirmed by FASTQC. RNA-seq reads were aligned to human reference genome GRCh38 assembly using HISAT2 (*Kim et al., 2015*) with alignment rate ~98% for all samples. Transcript assembly and expression assessment was performed by Stringtie (*Pertea et al., 2015*) to get the TPM (Transcripts Per Million) values for each gene. For direct visualization of RNA-seq signals, BigWig files were generated using deepTools with bamCoverage function, with RPKM normalization (*Ramírez et al., 2014*). Biological duplicates were merged via bigWigMerge (ucsc-bigwigmerge tools). Final bigwig files were visualized using both UCSC genome or Integrated Genome viewer (IGV).

Categorization of gene type was extracted from GRCh38 assembly GTF file downloaded from Ensemble (v94, from https://useast.ensembl.org/info/data/ftp/index.html). We summarized all types of pseudogenes into 'pseudogene' category. And 'others' refer to all the rest classes in our summary tables. The categories that were included in 'lncRNAs' in this study are described in *Supplementary file 1*: biotype_of_24087_genes.

For statistical analyses, raw gene counts were first analyzed by HTSEQ-Count (*Anders et al., 2015*), then analyzed using edgeR (*Robinson et al., 2010*). Qualifiable expression was defined by CPM >= 0.075 in at least two samples of total 10 samples. Normalization of library size was performed. For visualization of transcriptome, heatmaps were plotted using coolmap function from limma package (*Ritchie et al., 2015*), with row centering and scaling. Hierarchical clustering of genes (rows) was performed with complete-linkage method. Differential expression analyses were performed using exactTest between every two adjacent cell-cycle phases. Differentially expressed genes (DEGs) were defined by |fold change| >= 1.5 fold and FDR < 0.05. Phase-specific genes were further filtered from DEG lists by these criteria: (1) Genes show highest expression in that cell-cycle stage; (2) Significantly (FDR < 0.05) upregulated for >= 1.5 fold when compared to the two adjacent cell-cycle stages.

Gene ontology analyses (biological processes, Kegg pathway analyses) and GSEA (gene set enrichment analysis) were performed using clusterprofiler of Bioconductor (*Yu et al., 2012*). Specifically, gene ontology for biological process was performed using enrichGO function, Kegg pathway analyses was performed using enrichKEGG. All enrichment analyses include using background gene list containing all 24087 genes which showed qualifiable expression in the RNA-seq. Gene ontology networks results were visualized using Cytoscape. GSEA analysis was performed using gseGO function and gene lists were ranked using logFC values.

## siRNA treatment

SUNO1 siRNAs (listed in File 8) (Sigma) were transfected to cells, at a final concentration of 20 nM for twice (48 hr) with a gap of 24 hr, using Lipofectamine RNAiMax reagent (Invitrogen). Then cells were further cultured for another day before harvest. WTIP SMARTpool siRNAs were transfected to cells at a final concentration of 25 nM for twice. For the short-term *SUNO1* depletion, performed for the microarray analysis in *Figure 4A*, only one transfection of si*SUNO1* was applied, then cells were harvested 36 hr post transfection.

## Nuclear/cytoplasmic fractionation and chromatin fractionation

For nuclear and cytoplasmic fractionation, U2OS cells were lysed in lysis buffer (10 mM Tris-HCl (pH 7.4), 100 mM NaCl, 2.5 mM MgCl$_2$ and 40 µg/ml digitonin) by incubation on ice for 10 min. Nuclei were collected by centrifugation at 2,000 g at 4°C and lysed in Trizol reagent (Invitrogen). The supernatant was collected as the cytoplasmic fraction and mixed with Trizol LS reagent (Invitrogen) for RNA extraction.

For chromatin fractionation, U2OS cells were resuspended with solution A (10 mM HEPES pH7.9, 10 mM KCl, 1.5 mM MgCl$_2$, 0.34M sucrose, 1 mM DTT, 10% glycerol and 0.1% Triton X-100) and incubated on ice for 5 min. The cytoplasmic fraction (S2) was then separated from the nuclei by centrifuging at 4°C at 1,400 g for 4 min. Isolated nuclei were then washed with solution A without Triton X-100. The nuclei pellet was resuspended with solution B (3 mM EDTA, 0.2 mM EGTA, and 1 mM DTT) and incubated on ice for 30 min. The nuclear soluble fraction (S3) was then separated by centrifuging at 4°C at 1700xg for 4 min. The isolated chromatin fraction was then washed with buffer B. The chromatin pellet (P3) was resuspended in solution A and sonicated for 1 min.

## Single-molecule fluorescence RNA in-situ hybridization (smFISH)

The *SUNO1* smFISH probe set was designed using Stellaris Probe Designer (accession number AK124080.1), consisted of 32 20-mer DNA oligonucleotides. Oligonucleotides with a 3'amino group (LGC Biosearch Technologies) were pooled and coupled with Cy3 Mono NHS Ester (GE Healthcare).

HCT116 WT and *SUNO1* KO cells were seeded on coverslips coated with poly-L-lysine two days before experiments. At harvest, cells were fixed with freshly prepared fixative (3:1 Methanol-Glacial Acetic Acid) for 10 min at room temperature and washed with washing buffer (10% formamide, 2XSSC) for 5 min. Probe was added to hybridization buffer (10% dextran sulfate, 10% formamide in 2X SSC) at a final concentration of 125 nM. Hybridization was carried out as described in *Orjalo and Johansson, 2016* (*Orjalo and Johansson, 2016*) in a humidified chamber in the dark for 2 hr at 37°C. After hybridization, the coverslips were washed twice with wash buffer, 30 min for each wash, in the dark at 37°C. DNA was counterstained by DAPI during the second wash. The coverslips were then washed with 4XSSC for 5 min at room temperature and mounted in VectaShield Antifade Mounting Medium (Vector Laboratories). Images were taken using Zeiss Axiovert 200M microscope equipped with Cascade 512b high sensitivity camera.

## Northern blotting

Poly A+ RNA was fractionated from total RNA by NucleoTrap mRNA Mini Kit (Macherey-Nagel). 5 µg of Poly A+ RNA from HCT116 WT or KO cells were separated on 1% agarose gel prepared with NorthernMax Denaturing Gel Buffer (Ambion) and run in NorthernMax Running Buffer (Ambion). RNAs were then transferred to Amersham Hybond-N+ blot (GE Healthcare) by capillary transfer in 10 x SSC and crosslinked to the blot by UV (254 nm, 120mJ/cm$^2$).

The DNA probes were labeled with [$\alpha-32P$] dCTP by Prime-It II Random Primer Labeling Kit (Stratagene) as per manufacturer's instructions. Hybridization was carried out using ULTRAhyb Hybridization Buffer (Ambion) containing $1 \times 10^6$ cpm/ml of denatured radiolabeled probes overnight at 42°C. Blots were then washed with 2 x SSC, 0.1% SDS and 0.1 x SSC, 0.1% SDS sequentially at 42°C, and developed using phosphor-imager.

## Flow cytometry

For PI flow, cells were fixed by 90% chilled ethanol overnight. Fixed cells were washed and resuspended in PBS containing 1% NGS and then incubated with 10 µg/ml of RNase A and 120 µg/ml of propidium iodine (PI) for 30 min in the dark at 37°C. For BrdU-PI flow, cells were pulsed with 50 µM

BrdU for 30 min before collection. Cells were trypsinized, washed once in PBS, resuspended in 0.5 ml 0.9% NaCl and then added 0.5 ml chilled ethanol for fixation. After fixing overnight, cells were treated with 2N HCl/Triton X-100 solution for 25 min at room temperature to denature DNA. Cells were then washed once with 0.1M $Na_2B_4O_7$, resuspended in 1% BSA/PBS and stained with FITC-conjugated BrdU antibody (BD) for 1 hr. Cells were again washed and resuspended in PBS with 120 µg/ml propidium iodide (PI) and 10 µg/ml RNase A for 45 min in the dark at 37℃. Samples were analyzed on BD FACS Canto II analyzer. Data were processed using De Novo FCS Express five software.

## BrdU incorporation assay

For BrdU labeling, cells were incubated with 10 µM of BrdU for 20 min. Cells were then fixed with 2% PFA for 15 min at room temperature and permeabilized by 0.5% Triton X-100 for 10 min on ice. DNA was denatured by 4N HCl for 30 min at room temperature. Immunofluorescence staining of BrdU was performed using anti-BrdU antibody (Sigma) and anti-mouse Texas Red antibody. Images were taken using Axioimager.Z1 microscope (Zeiss) equipped with Hamamatsu ORCA-flash camera. Cells in S-phase (BrdU positive) were counted.

## Cell proliferation assay

HCT116 cells were incubated with control or SUNO1-specific siRNAs for 48 hr. After this, cells were reseeded into 6 cm plates at a density of $1.5 \times 10^5$ cells/plate. Cell numbers were then counted every 24 hr until day 5.

## Chromatin immunoprecipitation

Chromatin immunoprecipitation (ChIP) for DDX5 was performed using ChIP-IT High Sensitivity kit (Active Motif) according to manufacturer's protocol. 50 µg of cross-linked and sheared chromatin, and 5 µg of antibody were used for precipitation. Similarly, 5 µg of IgG was used to pull 50 µg of cross-linked and sheared chromatin as a control. Pol II ChIP was performed as reported earlier (*Khan et al., 2015*). Briefly, cells were fixed using freshly prepared 1% Formaldehyde solution for ten minutes at room temperature followed by quenching with 0.125 M Glycine solution. Cell were lysed, sonicated and precipitated using antibodies. 50 µg of cross-linked and sheared chromatin, and 5 µg of RNA Pol II antibody were used to pull the chromatin. 5 µg IgG was also used as a non-specificity control. qPCR was performed with purified DNA and results were analyzed as percent input.

## Immunoblotting

Cells were collected by scraping and lysed in lysis buffer containing protease inhibitors and phosphatase inhibitors for 10 min on ice. Loading dye was added to the lysate and samples were then heated at 95℃ for 5 min before loading onto a polyacrylamide gel. Western Blotting was performed as described previously (*Sun et al., 2018b*). Antibodies are listed in *Supplementary file 8*.

## Alkaline comet assay

Comet assay was performed using CometAssay Kit (Trevigen) following the manufacturer's instructions. Briefly, cells were collected by trypsinization, embedded in low-melting agarose and placed on CometSlides. After agarose solidifying, the slides were immersed in lysis solution for 30 min, incubated in alkaline unwinding solution then subjected to electrophoresis for 30 min. After washing in water and 70% ethanol for 5 min each, the slides were allowed to dry, and DNA was stained using SYBR safe.

## DNA fiber assay

Cells were labeled with 25 µM CldU for 30 min and then treated by 2 mM hydroxyurea for 24 hr followed by 30 min of labeling with 250 µM IdU. DNA fibers were prepared on vinyl-silane coated coverslips using the FiberComb molecular combing system (Genomic Vision) as per the manufacture's protocol. To visualize the CldU and IdU tracks, DNA fibers on coverslips were denatured in denaturation solution (0.5M NaOH, 1M NaCl) for 8 min at room temperature. Coverslips were then washed with PBS and dehydrated in 70%, 90%, and 100% ethanol for 5 min each. Coverslips were blocked with 1% BSA in PBST, followed by incubating in antibodies against CldU (anti-BrdU, 1:200, Bio-Rad,

OBT0030G) and IdU (anti-BrdU, 1:200, BD, 347580). After washing in BSA/PBST, the coverslips were incubated in FITC-conjugated goat anti-rat IgG and TexasRed-conjugated goat anti-mouse IgG. Images were taken using Axioimager.Z1 microscope (Zeiss) equipped with Hamamatsu ORCA-flash camera.

## Immunofluorescence staining

For YAP1 immunofluorescence staining coupled with EdU incorporation assay, cells were pulse-labeled by 10 µM EdU for 30 min and then fixed by 2% PFA for 15 min at room temperature. Cells were then permeabilized by 0.5% Triton X-100 for 10 min on ice. After washing with PBS, click reaction was performed with freshly prepared click cocktail (2 mM copper sulfate, 10 µM AF488-Azide, and 100 mM sodium ascorbate in PBS) for 1 hr at room temperature. Cells were then preceded to blocking step and YAP1 was stained by anti-YAP1 antibody (Santa Cruz) and Goat anti-Mouse AF568 antibody.

For immunostaining of DNA-damage markers, cells were pre-extracted by 0.5% Triton X-100 in cytoskeletal (CSK) buffer for 3 min on ice and then fixed by 2% PFA for 15 min at room temperature. Total RPA32 was stained by anti-RPA32 antibody (Cell signaling) and Goat anti-Rat TexasRed antibody. 53BP1 was stained by anti-53BP1 antibody (Cell signaling) and Goat anti-Rabbit Dylight 488 antibody. Images were taken using Axioimager.Z1 microscope (Zeiss) equipped with Zeiss AxioCam 506 Mono camera.

## Anchorage-dependent plastic colony formation assay

Cells were incubated with control or *SUNO1*-specific siRNAs for 48 hr. After that, cells were treated with DMSO (control) or Doxorubicin (300 nM) for 16 hr. After 16 hr, cells were washed with media to remove the drugs and were grown in fresh medium. Cells were reseeded in a 6-well plate at a density of 1000 cells per well. After 2 to 3 weeks, colonies were fixed with ice-cold 100% methanol for 5 min, stained with crystal violet and colonies were counted and analysis using ImageJ.

## Microarray analyses

Total RNA from control and *SUNO1*-depleted HCT116 cells were isolated using the RNeasy Plus Mini kit (Qiagen). 250 ng of total RNA was used for microarray analysis. Samples were labeled using the IlluminaTotalPrep RNA amplification kit (Ambion) according the instruction by the manufacture. 750 ng of cRNA was used for hybridization on microarrays using the HumanHT-12 v4 Expression BeadChip kit (Illumina) manufacturer's instructions and data was analyzed using the R/Bioconductor package (Bioconductor, Seattle, WA, USA). The microarray data are deposited in GEO with accession number GSE157393.

## Luciferase reporter assay

Wild-type CTGF promoter (WT) or TEAD-binding sites mutated CTGF promoter (mutant) luciferase reporters (kind gift from Dr. Kun-Liang Guan, UCSD) are transfected into control and *SUNO1*-depleted (si*SUNO1*-a or si*SUNO1*-b) U2OS cells, and 24 hr later, the relative luciferase activity is quantified using Dual-Luciferase Reporter Assay System (Promega, E1910) following the manufacturer's instructions.

## Chromosome conformation capture (3C) assay

The 3C assay was performed as described (*Dekker, 2006*), with minor modifications. Briefly, one million HCT116 were cross-linked with formaldehyde (final concentration 1%) for 15 min at room temperature, and resuspended in lysis buffer (10 mM Tris, pH 8.0, 10 mM NaCl, and 0.2% NP40) and incubated on ice for 90 min. One million of the prepared nuclei were digested with EcoRI (New England Biolabs) overnight at 37℃, followed by ligation with T4 DNA ligase (New England Biolabs) at 16℃ for 4 hr. The ligated DNA was incubated with Proteinase K at 65℃ for >8 hr or overnight to reverse the cross-links. Following incubation, the DNA was treated with RNase A. The treated DNA was extracted with phenol:chloroform and precipitated with sodium acetate (10% vol) and ethanol (2.5–3-fold volume). The DNA concentration of the recovered 3C library was determined using Qubit dsDNA HS assay kit (Invitrogen). Quantitative real-time PCR was performed to confirm the specific ligation between two DNA fragments - between *SUNO1* region and *WTIP* region, and between

*SUNO1* region and Control genomic region - in the sample libraries (*SUNO1* KD and *SUNO1* control) and BAC control libraries. Interaction frequencies were calculated by dividing the amount of PCR product obtained from the 3C sample library by the amount of PCR product obtained from the control library DNA generated from the corresponding BAC: Interaction frequency = 2(dCt sample – dCt control). The primers designed for 3C assay are: *SUNO1* region, 5'-TAGAACATGTTTCTTTG TCCAATAGGTGCTGAAAGGCCCG-3'; *WTIP* region, 5'- GGAGAGACGGGGTTTCACCATG TTGGCCAGGC-3'; and control region, 5'- ACCCCAGGCTCTCAGCAGCCGTGACCTCACAG-CACCAT-3'.

## RNA-affinity pulldown

RNA-affinity pulldown was performed as previously described (*Sun et al., 2018b*). Briefly, Biotin-labeled RNA probes were in vitro transcribed as per manufacturers' instructions (Biotin RNA labeling Mix, Roche; T7 polymerase, Promega) and purified by G-50 column (GE Healthcare). 2 μg purified biotinylated RNA was used for each pulldown.

Cells were resuspended in lysis buffer (10 mM Tris-HCl (pH 7.4), 100 mM NaCl, 2.5 mM MgCl$_2$, 40 μg/ml digitonin) and lysed on ice for 20 min with frequent mixing. Nuclei were then pelleted, resuspended in RIP buffer (150 mM KCl, 25 mM Tris pH 7.4, 0.5 mM DTT, 0.5% NP40), and sonicated three times for 5 s each. Debris were removed by centrifugation at 14,000 rpm for 10 min at 4℃. The nuclear lysate was then precleared by incubating with 40 μl of streptavidin beads (Dyna-beads M-280 streptavidin, Invitrogen) at 4℃ for 2 hr with rotation. The precleared lysate was incubated with the 2 μg biotinylated for 2 ~ 3 hr and then incubated with blocked beads at 4℃ overnight. Beads were then washed with high salt buffer (0.1% SDS, 1% Triton X-100, 2 mM EDTA, 20 mM Tris-HCl (pH 8.0), 500 mM NaCl), low salt buffer (0.1% SDS, 1% Triton X-100, 2 mM EDTA, 20 mM Tris-HCl (pH 8.0), 150 mM NaCl) and TE buffer. RNase Inhibitors, protease inhibitors, and phosphatase inhibitors were included in all the buffers used in the previous steps. Beads were then resuspended in SDS loading buffer and heated at 95℃ for 5 min. Protein samples were then analyzed by mass spectrometry or western blotting.

## DDX5 RNA-immunoprecipitation

HCT116 wild-type cells from two 10 cm plate were fixed with 1% formaldehyde in PBS, at room temperature for 10 min, and then Glycine was added at final concentration of 100 mM, and further incubated for 5 min. Following one wash with PBS, cells were then resuspended with 400 μl Lysis Buffer (1% SDS, 10 mM EDTA, 50 mM Tris-HCl pH8.1, supplemented with protease inhibitor cocktail and RNAase Inhibitor), and incubated at 4℃ for 30 min with rotation. The lysate was then sonicated with Bioruptor Diagenode (setting 'High', 15 mins, three times). Centrifuge the sonicated lysate at 10, 000 rpm for 5 min to remove the debris. Then transfer the supernatant, make up volume to 1 ml with IP buffer (0.01% SDS, 1.1% Triton X-100, 1.2 mM EDTA, 16.7 mM Tris-HCl, pH 8.1, 167 mM NaCl), add to 50 μl of pre-washed Gamma Bind G Sepharose beads, incubated at 4℃ for 1 hr with rotation, to pre-clear the lysate. After pre-clearing, centrifuge at 10,000 rpm for 5 min, transfer the supernatant. Keep 100 μl as input, 450 μl for IgG, and 450 μl for IP with mouse monoclonal anti-DDX5 antibody (Millipore, Cat#: 05–580). Incubate at 4℃ overnight with rotation. On the next day, add pre-washed Sepharose beads, incubate at 4℃ for 2 hr with rotation. Then wash the beads once with IP buffer, once with High Salt Buffer (0.1% SDS, 1% Triton X-100, 2 mM EDTA, 20 mM Tris-HCl pH 8.1, 500 mM NaCl), once with TE buffer, 5 min rotation at 4℃ for each wash. Elute in 165 μl Elution Buffer (1% SDS, 0.1M NaHCO3, RNAase Inhibitor), incubate at 37℃ for 15 min, repeat once, combine the elute. To 330 μl eluate, add 14 μl 5M NaCl. To the Input sample, add 282 μl Elution buffer and 14 μl 5M NaCl. Incubate at 65℃ with vortex for 2 hr. Add 1032 μl Trizol LS (Invitrogen) to IP/IgG/Input sample. Proceed with RNA isolation following manufacturer's instructions.

## Nascent RNA capture assay

Nascent RNAs were labeled and captured using Click-iT Nascent RNA capture kit (Life Technologies) as per the manufacturer's instructions. Expression level of nascent RNAs were quantified by qRT-PCR.

## Tumor xenograft assay

Immunocompromised mice ($neu^{-/-}$) were obtained from Jackson laboratory (females) and used for the xenograft experiment. A cohort of 5 mice were used for this study. The mice were injected with one million control HCT116 cells on the right flank and equal number of *SUNO1* KO HCT116 cells were injected into the left flank of the same cohort of mice. Tumors were measured with a digital caliper (length (mm) X breadth (mm) X height (mm)) every five days. The graph denotes the mean of five tumor volume for each cell line.

## Data analyses and statistics

Relative RNA levels were normalized to GAPDH or 18S RNA. Results are represented as mean ± SD of three independent experiments. Two-tailed Student's t-tests were performed. $*p < 0.05$, $**p < 0.01$, $***p < 0.001$.

## PCR primers, qPCR Primer, siRNA, and gRNA sequences

See *Supplementary file 8* for the details.

## Antibodies

See *Supplementary file 8* for the details.

# Acknowledgements

We thank members of Prasanth's laboratory for their valuable comments. We thank Drs. Sayee Anak (UIUC) (YAP antibody), Erik Bolton (UIUC) (Cyclin D1 antibody), Kun-Liang Guan (UCSD) (CTGF-promoter reporter constructs), Dr. Kenneth Irvine (Rutgers University (pTRIPZ-EGFP:*WTIP*)) for providing reagents, and Dr. Alvaro G Hernandez (UIUC Genomic facility) for RNA-sequencing. We also thank Dr. Jian Ma, Dr. Yang Zhang and Omid Gholamalamdari for technical discussion relating to bioinformatic analyses. We thank Jon Zetterval for his assistance on the Xenograft experiments. This work was supported by National Institute of Health [R01GM088252, R01GM132458 and R21AG065748 to KVP, GM125196 to SGP and GM123314 to SCJ], Cancer center at Illinois seed grant and Prairie Dragon Paddlers to KVP, National Science Foundation [EAGER grant to KVP {1723008} and career award {1243372} and 1818286 to SGP]. AL was supported by the Intramural Research Program of the National Cancer Institute (NCI), Center for Cancer Research (CCR). Research in the SD lab is supported by Deutsche Forschungsgemeinschaft (Di 1421/7–1) and Deutsche Krebshilfe.

# Additional information

### Competing interests

Ashish Lal: Reviewing editor, *eLife*. Arturo V Orjalo: Arturo V Orjalo is affiliated with LGC Biosearch Technologies and Genentech Inc, the author has no financial interests to declare. Hans E Johansson: Hans Johansson is affiliated with LGC Biosearch Technologies. The author has no financial interests to declare. The other authors declare that no competing interests exist.

### Funding

| Funder | Grant reference number | Author |
| --- | --- | --- |
| National Institute of General Medical Sciences | GM088252 | Kannanganattu V Prasanth |
| National Institute of General Medical Sciences | GM125196 | Supriya G Prasanth |
| National Institute of General Medical Sciences | GM123314 | Sarath C Janga |
| National Science Foundation | 1723008 | Kannanganattu V Prasanth |
| National Science Foundation | 1243372 | Supriya G Prasanth |
| National Institute on Aging | AG065748 | Kannanganattu V Prasanth |

| American Cancer Society | RSG-11-174-01-RMC | Kannanganattu V Prasanth |
| National Institute of General Medical Sciences | R01GM132458 | Kannanganattu V Prasanth |
| National Science Foundation | 1818286 | Supriya G Prasanth |

The funders had no role in study design, data collection and interpretation, or the decision to submit the work for publication.

## Author contributions

Qinyu Hao, Conceptualization, Data curation, Formal analysis, Validation, Investigation, Visualization, Methodology, Writing - original draft, Writing - review and editing; Xinying Zong, Conceptualization, Data curation, Formal analysis, Supervision, Funding acquisition, Validation, Investigation, Visualization, Methodology, Writing - original draft, Project administration, Writing - review and editing; Qinyu Sun, Yo-Chuen Lin, Resources, Data curation, Software, Formal analysis, Validation, Investigation, Visualization, Methodology, Writing - review and editing; You Jin Song, Yuelin J Zhu, Data curation, Formal analysis; Seyedsasan Hashemikhabir, Data curation, Software, Formal analysis, Investigation, Visualization, Methodology, Writing - review and editing; Rosaline YC Hsu, Formal analysis, Investigation, Methodology; Mohammad Kamran, Data curation, Software, Formal analysis, Validation, Investigation, Methodology, Writing - review and editing; Ritu Chaudhary, Vidisha Tripathi, Deepak Kumar Singh, Data curation, Formal analysis, Validation, Investigation, Methodology, Writing - review and editing; Arindam Chakraborty, Maria Polycarpou-Schwarz, Validation, Investigation, Methodology, Writing - review and editing; Xiao Ling Li, Yoon Jung Kim, Formal analysis, Investigation, Methodology, Writing - review and editing; Arturo V Orjalo, Formal analysis, Validation, Investigation, Methodology, Writing - review and editing; Branden S Moriarity, Resources, Investigation, Methodology, Writing - review and editing; Lisa M Jenkins, Resources, Methodology; Hans E Johansson, Resources, Supervision, Investigation, Methodology; Sven Diederichs, Conceptualization, Resources, Supervision, Investigation, Methodology; Anindya Bagchi, Conceptualization, Supervision, Investigation, Methodology; Tae Hoon Kim, Supervision, Investigation, Methodology; Sarath C Janga, Resources, Data curation, Software, Supervision, Investigation, Methodology; Ashish Lal, Conceptualization, Formal analysis, Supervision, Investigation, Methodology, Writing - review and editing; Supriya G Prasanth, Conceptualization, Resources, Data curation, Software, Supervision, Investigation, Methodology, Writing - review and editing; Kannanganattu V Prasanth, Conceptualization, Formal analysis, Supervision, Funding acquisition, Investigation, Methodology, Project administration, Writing - review and editing

## Author ORCIDs

Qinyu Hao (ID) https://orcid.org/0000-0002-7059-7741
Mohammad Kamran (ID) http://orcid.org/0000-0003-4210-672X
Ashish Lal (ID) http://orcid.org/0000-0002-4299-8177
Supriya G Prasanth (ID) https://orcid.org/0000-0002-3735-7498
Kannanganattu V Prasanth (ID) https://orcid.org/0000-0003-4587-8362

## Decision letter and Author response

Decision letter https://doi.org/10.7554/eLife.55102.sa1
Author response https://doi.org/10.7554/eLife.55102.sa2

## Additional files

### Supplementary files

• Supplementary file 1. Gene count, gene expression (TPM), and biotype of quantifiable genes of RNA-seq. First sheet 'gene_count_all' contains raw counts from HTSeq-count analysis. Second sheet 'TPM_all' contains TPM (Transcripts Per Million) as the expression level of each gene in all samples. TPM is calculated using Stringtie. Third sheet 'list_of_24087_genes' includes the genes that have quantifiable expression (CPM >= 0.075 in at least two samples). Last sheet

'biotype_of_24087_genes' includes the detailed categorization information of these genes. The bio-type information is based on Ensemble (https://useast.ensembl.org/info/genome/genebuild/biotypes.html).

• Supplementary file 2. Differential expression results. Five sheets represent the full results (of 24087 genes) of differential expression tests (exactTest from edgeR) between G1 vs. G1S, G1S vs. S, S vs. G2, G2 vs. M, M vs. G1, respectively.

• Supplementary file 3. DEG list and biotype classification. File three is a subset of File two and it includes only DEGs information. The gene categories are also provided as individual sheets. Statistics summarizing the categorization of each comparison (between two phases) is listed in *Figure 1B*.

• Supplementary file 4. Gene ontology and GSEA. Six sheets represent the detailed, full output from GSEA/GO/Kegg pathway analyses in this study. They correspond to data presented in *Figure 1—figure supplement 2A*, *Figure 1—figure supplement 2B*, *Figure 1D*, *Figure 1—figure supplement 2C*, respectively.

• Supplementary file 5. Phase-specific genes. First sheet 'all_phase_specific_with_TPM' includes all 5162 phase-specific genes and their TPM values. Second sheet '2044_phase_specific_lncRNAs' shows the list of 2044 lncRNAs only. The lncRNA categorization criteria are explained in detail in *Supplementary file 1*, last sheet, 'biotype_of_24087_genes'. Statistics summarizing the categorization is listed in *Figure 2—figure supplement 1A*.

• Supplementary file 6. Deregulated genes in *SUNO1* KD cells compared to control cells detected by Microarray analyses.

• Supplementary file 7. *SUNO1*-binding proteins detected by RNA affinity Pulldown followed by Mass Spectrometry.

• Supplementary file 8. List of primers, siRNAs, gRNAs, and antibodies.

• Transparent reporting form

## Data availability

Sequencing data have been deposited in GEO under accession code GSE143275. Microarray data has been deposited in GEO under the accession number GSE157393.

The following datasets were generated:

| Author(s) | Year | Dataset title | Dataset URL | Database and Identifier |
|---|---|---|---|---|
| Hao Q, Prasanth KV, Sun Q | 2020 | poly A+ RNA sequencing of cell cycle-synchronized RNA from U2OS cells | https://www.ncbi.nlm.nih.gov/geo/query/acc.cgi?acc=GSE143275 | NCBI Gene Expression Omnibus, GSE143275 |
| Hao Q, Prasanth KV, Sun Q | 2020 | Microarray analyses to determine deregulated genes in *SUNO1* KD cells compared to control cells | https://www.ncbi.nlm.nih.gov/geo/query/acc.cgi?acc=GSE157393 | NCBI Gene Expression Omnibus, GSE157393 |

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
