## [Decision Letter]

**Acceptance summary:**

This study identifies the lncRNA *SUNO1* as an important mediator of cell cycle progression. *SUNO1* is expressed during S phase of the cell cycle and promotes expression of *WTIP*, a positive regulator of the transcription factor YAP1. Importantly, dysregulated *SUNO1* expression is associated with poor cancer prognosis.

**Decision letter after peer review:**

Thank you for submitting your article "The S phase-induced *SUNO1* lncRNA promotes cell proliferation by controlling YAP1/Hippo signaling pathway" for consideration by *eLife*. Your article has been reviewed by three peer reviewers, one of whom is a member of our Board of Reviewing Editors, and the evaluation has been overseen by Kevin Struhl as the Senior Editor. The reviewers have opted to remain anonymous.

The reviewers have discussed the reviews with one another and the Reviewing Editor has drafted this decision to help you prepare a revised submission.

Summary:

This is a study of cell cycle-dependent lncRNAs. One S-phase lncRNA (*SUNO1*) was chosen for further analysis. The authors conclude the *SUNO1* is required for S phase entry and that *SUNO1* deficient cells are more sensitive to DNA damage. The authors conclude that *SUNO1*, by modulating the functional interaction between DDX5 and RNA pol II, regulates pro-survival pathways such as Hippo/YAP1 cell signaling thereby affecting cell proliferation. These are interesting conclusions, but the data provided fall short of fully validating these conclusions.

Essential revisions:

1) The authors did not perform qPCR-based validation for the obtained RNA-seq data. Among thousands of identified lncRNAs in this study, the authors also did not rationalize the selection of S phase specific lncRNA *SUNO1* given its very low abundancy as evidenced by the RNA-FISH. The quality of the northern blot presented in Figure 2—figure supplement 3C is very poor and there is a possibility of DNA cross-contamination induced by the DNA probes. The authors do not provide details on the probes used in the northern blotting. Moreover, the northern blotting suggests the presence of two distinct isoforms. *SUNO1*, according to hg38, is a monoexonic, measuring 2040 bp. However, according to Figure 2A, read distribution is very uneven across 2 kb region, including S-phase. In particular, the read distribution in S phase exceeds more than 2 kb, covering close to 3 kb region. However, the authors predict two bands at 2 kb and 5 kb in northern blots provided in Figure 2G. Thus, the bands in northern blots are not consistent with the RNA-seq data. The lack of proper isoform characterization is compounded by the absence of knowledge concerning which isoform they are investigating or silencing.

2) Based on Figure 2C, the authors suggest that nuclear puncta represent *SUNO1*, and they justify this due to lack of signal in CRISPR/Cas9 KO cells. However, according to Figure 3—figure supplement 1E CRISPR deletion spans more than 1.5 kb of 2 kb *SUNO1* transcriptional unit. Considering the extent of SUNO deletion, the authors should explain the location of probe (details in the Materials and methods section are not apparent) used for northern blotting, and also probes used for RNA-FISH.

3) The CRISPR-mediated *SUNO1* KO cells express Cas9. However, it is not clear that the WT cells express Cas9. Is it possible that the PI FACS phenotype is caused by Cas9-induced DNA damage? It is stated in many places in the text that *SUNO1* is required for S-phase entry. If so, what explains the apparent S-phase population of *SUNO1* depleted and SUNO KO cells? The reported proliferation assays are restricted to PI FACS assays. These studies should be complemented by measurement of cell number over time and by studies of DNA synthesis (e.g. EdU incorporation). Does expression of *SUNO1* have "gain-of-function" phenotype in cell cycle regulation, as well as downstream target transcription? Does *SUNO1* expression rescue the proliferation phenotype of *SUNO1* knockout cells?

4) The intrinsic DNA damage caused by *SUNO1* depletion alone is supported by a DNA comet assay (Figure 3—figure supplement 2A) and to a certain extent by p53 activation (Supplementary Figure 3H). The authors should provide γH2AX Western for *SUNO1* depleted cells following siRNA and CRISPR/Cas9 clones. Does expression of *SUNO1* rescue the DNA damage phenotype in CRISPR KO clones?

5) The authors refer to the syntenic homology with the mouse *SUNO1* without characterizing or providing further information on the mouse lncRNA. Authors should show that mouse *SUNO1* rescues the functions of its human homolog. Otherwise, the current data on the mouse homolog becomes irrelevant to the manuscript.

6) The DNA fiber assay experiment was used to investigate the effect of *SUNO1* in combination with HU treatment. However, this technique should be used to examine the role of *SUNO1* in DNA replication (not solely post-HU treatment). The authors conclude that *SUNO1* modulation affects origin licensing – thus conclusion needs to be supported by examining status of pre-replication and replication complexes such as ORC and MCMs loading.

7) The interpretation of the HU study presented in Figure 3 is confusing. Since HU causes replication fork collapse and *SUNO1* is required for S phase entry, HU would not be expected to cause replication fork collapse in *SUNO1* knockout cells. How do these experiments demonstrate a role for *SUNO1* in a DNA damage check point?

8) The 3C experiment (Figure 5A) lacks control. Primers spanning *SUNO1* and *WTIP* are used. Primers spanning intervening genes (which seem not controlled by *SUNO1* as negative controls) are also needed. Moreover the experiment shown also indicates that the *SUNO1* genomic locus may be more crucial to the physical association with *WTIP* promoter than the transcript per se. Do the authors rule out the act of *SUNO1* transcription in the regulation *WTIP*?

9) The *WTIP* studies are poorly described. Why were the microarray studies done using *SUNO1* siRNA rather than *SUNO1* KO cells? What time point is the immunoblot (Figure 4C). There appears to be no description of siRNA sequences. It appears that two siRNA were used in Figure 4—figure supplement 1D, but the sequences and time are not described. Only a single siRNA was used to deplete *WTIP* (Figure 4—figure supplement 1G). The peak of *SUNO1* expression is in S phase (Figure 2—figure supplement 1C) while *WTIP* peaks in G1/S (Figure 4—figure supplement 1F) – if *SUNO1* increases *WTIP* expression, what accounts for this? The PI FACS analysis of siWTIP (Supplementary Figure 4D) is not sufficient to support the conclusions presented (see point 3, above).

10) Figure 4C shows that si*SUNO1* decreases YAP expression – what is the mechanism? This decrease could be sufficient to explain the loss of YAP transcriptional activity. Does *SUNO1* affect YAP localization? Is this caused by *SUNO1* deficiency or *WTIP* deficiency – can be tested by complementation analysis?

11) The authors should strengthen the functional connection between *SUNO1* and WTIP/YAP1. Does WTIP or YAP1 rescue *SUNO1* mediated functions? The significance of interaction between *SUNO1* and DDX5 in target gene regulation is unclear if DDX5 recruitment to target gene promoters is not dependent on *SUNO1*. How does *SUNO1* affects the interaction between DDX5 and RNA pol II?

12) The authors should provide further evidence to strengthen the connection to Hippo signaling. For example, does *SUNO1* deletion affect YAP cellular localization, in addition to its expression level? Does it also affect the TAZ level? Expression of other bona fide YAP/TAZ target genes, such as *CYR61* and ANKRD1, should be included in the analysis. In addition, they should examine whether *SUNO1* affects phosphorylation of the upstream Hippo kinases, Mst and LATS.

13) It was recently reported that expression of the components of the Hippo pathway, including YAP/TAZ, oscillate during mitotic cell cycle (Kim et al., 2019). Thus, it is important to test whether cell cycle regulation of YAP/TAZ is indeed dependent on *SUNO1* or WTIP in this context.

14) The authors state that "Interestingly, a major fraction of these genes (71%), including WTIP, showed a positive correlation in expression with *SUNO1* across colon cancer patients" However the statistical significance of the correlated genes is not provided.

15) The soft agar assays cannot be interpreted without studies of 2D cultures. Is the loss of growth in agar simply because the *SUNO1* KO cells do not grow?

16) Many experiments use a single siRNA. At least two siRNAs or siRNA/CRISPR strategies should be used to strengthen their conclusions. In addition to qRT-PCR validation of target genes, Western blot analyses should be included.

---

## [Author Response]

Essential revisions:1) The authors did not perform qPCR-based validation for the obtained RNA-seq data.

We apologize for not presenting the RT-qPCR validation data of other lncRNA candidate genes in the earlier version of the manuscript. We have now included RT-qPCR validation of nine candidate lncRNAs that shows elevated expression during S phase in the RNA-seq data. (Figure 2—figure supplement 1B).

Among thousands of identified lncRNAs in this study, the authors also did not rationalize the selection of S phase specific lncRNA SUNO1 given its very low abundancy as evidenced by the RNA-FISH.

Again, we apologize for not explaining the rationale of selecting *SUNO1* for functional studies in the manuscript. We selected multiple S phase upregulated lncRNAs for mechanistic experiments using the following criteria.

a) Significantly elevated expression during S phase.

b) Distinct chromatin marks (H3K27 acetylation, H3K4 trimethylation) in regulatory elements of the lncRNA genes.

c) Depletion of lncRNA utilizing 2-3 independent siRNA/shRNAs in multiple cell lines showing consistent cell cycle defect phenotypes.

*SUNO1* satisfied all of these criteria. In addition, *SUNO1* shows S phase expression in multiple cell lines, based on studies from our laboratory in HeLa as well as recent independent RNA-seq studies from other laboratories in MCF-7 (Liu et al., 2017) and hTERT-RPE1 (Yildirim et al., 2020) (Figure 2—figure supplement 1D-F). We agree with the reviewer that it is difficult to study the function of low abundant lncRNAs. However, most of the lncRNAs in cells are expressed with low abundance (in comparison to protein-coding genes) and display cell type-specific expression. Several earlier studies have demonstrated the involvement of low copy lncRNAs, such as *lincRNA-p21* in cell cycle functions (Dimitrova et al., 2014; Seiler et al., 2017). As a matter of fact, lncRNAs that regulate gene expression in *cis* tend to express in low copy numbers (Dimitrova et al., 2014).

The quality of the northern blot presented in Figure 2—figure supplement 3G is very poor and there is a possibility of DNA cross-contamination induced by the DNA probes.

Weak *SUNO1* northern blot (NB) bands observed in HCT116 poly A^+^ RNA was because of its low abundance. However, lack of the specific signal in the *SUNO1* KO cells confirms the specificity of the signal observed in the wild type cells (please also see the response to critique 1-5). To further confirm the specificity of the bands, we performed NB in BT20 human breast cancer cells (Figure 2—figure supplement 3B). BT20 cells showed >70 fold of *SUNO1* expression compared to HCT116 cells (Figure 2—figure supplement 1G). Based on the NB data in BT20 cells, it is evident that >2.1 kb isoform of *SUNO1* represents the major isoform. In addition, *SUNO1* is also presented as a low abundant long isoform (>5 kb) in both HCT116 and BT-20 cell lines.

The authors do not provide details on the probes used in the northern blotting.

We apologize for not providing this essential information in the manuscript. We have now included the following information in the revised manuscript (Figure 2—figure supplement 3A).

a) The probe used in the northern blotting.

b) The probe used in smFISH.

c) The region deleted in the CRISPR KO cells.

d) Positions of the siRNAs.

e) UCSC tracks of BT-20 RNA-seq datasets.

Moreover, the northern blotting suggests the presence of two distinct isoforms. SUNO1, according to hg38, is a monoexonic, measuring 2040 bp. However, according to Figure 2A, read distribution is very uneven across 2 kb region, including S-phase. In particular, the read distribution in S phase exceeds more than 2 kb, covering close to 3 kb region. However, the authors predict two bands at 2 kb and 5 kb in northern blots provided in Figure 2G. Thus, the bands in northern blots are not consistent with the RNA-seq data. The lack of proper isoform characterization is compounded by the absence of knowledge concerning which isoform they are investigating or silencing.

The uneven read distribution observed in cell cycle-synchronized RNA-seq from U2OS cells is due to the low abundance of *SUNO1*. However, the RNA-seq data from BT-20 cells (Ghandi et al., 2019; Varley et al., 2014) clearly demonstrated enhanced reads within an ~2 kb region, which matched with the annotated *SUNO1* region (Figure 2—figure supplement 3A). We named this isoform *SUNO1*-s. In addition, we observed reads expanding through a >5 kb long region downstream of the CpG island. We named this long isoform *SUNO1*-l. *SUNO1*-l shows much less expression level compared to *SUNO1*-s. Both of the bands are labeled in the revised manuscript Figure 2—figure supplement 3B-C.

Moreover, we analyzed the publicly available GRO-seq datasets in HCT116 and MCF7 cells (Andrysik et al., 2017) (Figure 2—figure supplement 3D). We observed a distinct transcription starting from the annotated TSS of *SUNO1*. Because of the bidirectional feature of the CpG island, there is transcription activity on the opposite strand of *SUNO1* but the RNA produced from the reverse strand shows a much lower level than *SUNO1*. The CAGE as well as poly A-seq datasets further confirm the 5’ and 3’ of the *SUNO1* transcript (Figure 2—figure supplement 3D).

2) Based on Figure 2C, the authors suggest that nuclear puncta represent SUNO1, and they justify this due to lack of signal in CRISPR/Cas9 KO cells. However, according to Figure 3—figure supplement 1E CRISPR deletion spans more than 1.5 kb of 2 kb SUNO1 transcriptional unit. Considering the extent of SUNO deletion, the authors should explain the location of probe (details in the Materials and methods section are not apparent) used for northern blotting, and also probes used for RNA-FISH.

We apologize for not providing this critical information. We have now included this information in the revised manuscript (Figure 2—figure supplement 3A).

Please note that the DNA region corresponding to the probe that was used in NB to detect *SUNO1*, (located within the 3’end of *SUNO1* gene) is *intact* even in the CRISPR-KO cells (Figure 2—figure supplement 3A). The lack of bands (2kb and >5 kb) in the NB of *SUNO1* KO cells indicates that *SUNO1* probe specifically detects only the *SUNO1* isoforms.

3) The CRISPR-mediated SUNO1 KO cells express Cas9. However, it is not clear that the WT cells express Cas9. Is it possible that the PI FACS phenotype is caused by Cas9-induced DNA damage?

The control (WT) cells that were used to compare with CRISPR KO cells were also generated by transfecting all of the plasmids (including Cas9) except the gRNAs followed by the same drug selection procedure as of the KO clones. Cas9 was transiently transfected into the WT and KO cells. We have now included this information in the manuscript. We therefore believe that the specific DNA damage defect observed in the *SUNO1* knock down cells are unlikely due to the nonspecific activity of transiently expressed Cas9. Furthermore, the DNA damage phenotype is also observed upon depletion of *SUNO1* using siRNA approach.

It is stated in many places in the text that SUNO1 is required for S-phase entry. If so, what explains the apparent S-phase population of SUNO1 depleted and SUNO KO cells?

S-phase entry is a critical biological process that is regulated tightly by several parallel gene regulatory mechanisms (E2F-, YAP/TEAD-, Myc-, FOS/Jun-mediated transcription). We argue that *SUNO1*-regulated YAP signaling is one of the mechanisms that control the expression of genes required for cell cycle S phase progression. We observed a decrease in the S-phase population in *SUNO1* KO cells and a concomitant accumulation in G1 phase, supporting our argument. This type of parallel gene regulatory processes is not unique to lncRNAs. For example, depletion of key pre-replication proteins (including Orc2 and ORCA/LRWD1) in cancer cells, causes the cells to accumulate in G1 with fewer cells progressing into S-phase of the cell cycle, consistent with licensing of fewer origins (Prasanth, Prasanth, and Stillman, 2002; Shen et al., 2010).

The reported proliferation assays are restricted to PI FACS assays. These studies should be complemented by measurement of cell number over time and by studies of DNA synthesis (e.g. EdU incorporation).

We thank the reviewer/s for their suggestion. We have now performed multiple assays to confirm the S phase progression and proliferation defects observed upon SUNO depletion or deletion. We performed BrdU incorporation assay (Figure 3—figure supplement 1D), EdU incorporation assay (Figure 4D) to demonstrate decrease in S phase cells after *SUNO1* depletion. In addition, BrdU-PI-flow cytometry (Figure 3B) analyses also confirmed decrease of S-phase population with a concomitant increase in G1 population in cells depleted of *SUNO1*. Finally, the growth curve assays showed defects in cell proliferation in *SUNO1*-depleted cells (Figure 3C).

In addition, we also performed BrdU-PI flow (Figure 3—figure supplement 1G) as well as growth curve analyses (Figure 3—figure supplement 1H) in *SUNO1*-knock out (KO) cells. *SUNO1*-KO cells also showed significant reduction of S-phase population and slower growth.

Does expression of SUNO1 have "gain-of-function" phenotype in cell cycle regulation, as well as downstream target transcription? Does SUNO1 expression rescue the proliferation phenotype of SUNO1 knockout cells?

Transient overexpression of *SUNO1* using plasmid-based transient transfection did not rescue the cell cycle phenotype observed upon *SUNO1* depletion. Our data indicate that *SUNO1* gene and/or RNA primarily functions in *cis*. The proximal location of *SUNO1* gene as well as the association of *SUNO1* transcripts at the *WTIP* genomic locus may be crucial for its function. Thus, exogenously expressing *SUNO1* may not be able to rescue the proliferation phenotype of *SUNO1*-depleted cells.

4) The intrinsic DNA damage caused by SUNO1 depletion alone is supported by a DNA comet assay (Figure 3—figure supplement 2A) and to a certain extent by p53 activation (Supplementary Figure 3H). The authors should provide γH2AX Western for SUNO1 depleted cells following siRNA and CRISPR/Cas9 clones. Does expression of SUNO1 rescue the DNA damage phenotype in CRISPR KO clones?

We respectfully point out that the DNA comet assay is one of the most-accepted assays to directly quantify DNA damage. It is considered a gold-standard method to assess DNA damage. Increased expression or differential localization of DNA damage marker proteins is used as an indirect measure to identify DNA damage.

As per the reviewer's suggestion, we looked at the nuclear foci formation of several DNA damage markers in control and *SUNO1*-depleted asynchronous cells. We observed increased levels of RPA32 and 53BP1 nuclear foci in *SUNO1*-depleted cells, further supporting DNA damage (Figure 3—figure supplement 2B).

In addition, we also observed small but consistent increase of pChk2 level upon *SUNO1*-depletion, implying activation of ATM (Figure 3—figure supplement 2C).

We previously demonstrated that *SUNO1* depleted cells showed reduced gH2AX even after DNA damage (Figure 3—figure supplement 3B). We also observed reduced phosphorylation of ATR substrates (pChk1, pBRCA1, pRPA32) in HU-treated *SUNO1*-depleted cells supporting our model that ATR signaling is defective in these cells.

We believe that in addition to increased DNA comet activity, enhanced levels of p53 protein (Figure 3—figure supplement 2C) and p21 mRNA (Figure 5G), increased RPA32 and 53BP1 foci (Figure 3—figure supplement 2B) and pChk2 (Figure 3—figure supplement 2C) observed in *SUNO1*-depleted cells are indicative of DNA damage.

5) The authors refer to the syntenic homology with the mouse SUNO1 without characterizing or providing further information on the mouse lncRNA. Authors should show that mouse SUNO1 rescues the functions of its human homolog. Otherwise, the current data on the mouse homolog becomes irrelevant to the manuscript.

We agree with the reviewer that we have not characterized the mouse ortholog of *SUNO1*. As per the suggestion, we have now removed the mouse *SUNO1* data from the manuscript, which will be followed in future investigations.

6) The DNA fiber assay experiment was used to investigate the effect of SUNO1 in combination with HU treatment. However, this technique should be used to examine the role of SUNO1 in DNA replication (not solely post-HU treatment).

We thank this reviewer for his/her suggestion. We have now performed DNA fiber assay in asynchronous control and *SUNO1*-depleted HCT116 cells (Author response image 1). Our results suggest that once the replication initiates, the fork progresses at a normal pace in *SUNO1*-depleted cells. We do observe that fewer cells are in S-phase in *SUNO1*-depleted cells and in these cells we observe a significant decrease in origin density.

**Author response image 1. sa2fig1:** Control and *SUNO1*-depleted cells were incubated with CldU (30min), washed using the media followed by IdU (another 30 min). DNA fibers were made (DNA combing), were stained using antibodies to detect CldU- and IdU-incorporated DNA fibers. a) Fiber length were quantified from microscopic images. n=120 fibers. Unpaired two-tail t-tests are performed. ns, not significant, *p* > 0.05. b) Firing events are calculated by counting the number of fibers containing an origin over the total fibers. Data was presented as Mean ± SD, n=3. Unpaired two-tail t-tests are performed. ***p* < 0.01.

The authors conclude that SUNO1 modulation affects origin licensing – thus conclusion needs to be supported by examining status of pre-replication and replication complexes such as ORC and MCMs loading.

We thank this reviewer for his/her comments. We tested the chromatin loading of core MCM (MCM3) and ORC (Orc2) components in presence or absence of *SUNO1* (Figure 3E). We observed defects in the chromatin loading of MCM complex (and not ORC) upon *SUNO1* depletion. This is consistent with fewer licensing origins and therefore accumulation of cells in G1 phase. Some cells that do enter S-phase with fewer licensed origins have normal fork progression, but longer S-phase.

7) The interpretation of the HU study presented in Figure 3 is confusing. Since HU causes replication fork collapse and SUNO1 is required for S phase entry, HU would not be expected to cause replication fork collapse in SUNO1 knockout cells. How do these experiments demonstrate a role for SUNO1 in a DNA damage check point?

*SUNO1* depletion led to a significant reduction but NOT complete loss of S-phase population. Furthermore, our experiments revealed that *SUNO1*-depleted cells showed slow S phase progression post HU release compared to control cells. Our results show that in the *SUNO1*-depleted cells, not only there are fewer licensed origins, but also a defect in firing of dormant origins in the vicinity of stalled forks. Dormant origins and the S-phase checkpoint are critical for rescuing stalled forks and therefore needed for the completion of DNA replication under replication stress (Yekezare, Gomez-Gonzalez, and Diffley, 2013).

8) The 3C experiment (Figure 5A) lacks control. Primers spanning SUNO1 and WTIP are used. Primers spanning intervening genes (which seem not controlled by SUNO1 as negative controls) are also needed.

We thank the reviewer for his/her suggestions. Unfortunately, we could not do the experiment suggested by the reviewer. The collaborator (Prof. Tae Hoon Kim, UT Dallas) who performed the 3C analyses in the manuscript has closed down his lab in order to take up an administration position. Due to the Covid19 situation, we could not initiate a new collaboration with another lab to do the 3C analyses.

We looked at the ENCODE Hi-C datasets to see potential interactions between *WTIP* and *SUNO1* genes. We observed that both the genes (and several genes that are located in proximity) are part of the same TAD based on Hi-C data of multiple cell lines including HCT116 (Rao et al., 2017) (Figure 5—figure supplement 1). Thus, the public Hi-C data supports our 3C results. The negative control primer amplifies a genic region that is located ~156 kb upstream of *SUNO1*. These two genomic regions are not part of the same TAD (Figure 5—figure supplement 1) and as expected, they showed no interaction by 3-C (Figure 5A).

Moreover the experiment shown also indicates that the SUNO1 genomic locus may be more crucial to the physical association with WTIP promoter than the transcript per se. Do the authors rule out the act of SUNO1 transcription in the regulation WTIP?

We thank the reviewer for this comment. It has been very difficult to separate the role of transcription/chromatin versus transcript when it comes to characterizing lncRNA-mediated *cis*-gene regulation. 3C analyses indicated that *SUNO1* gene locus is located in close proximity to *WTIP* locus even in the presence or absence of *SUNO1* lncRNA. We interpret that proximity-association between *SUNO1* and *WTIP* genes helps the low copy number *SUNO1* RNA to associate with the regulatory elements within the *WTIP* gene locus, to potentially recruit and/or stabilize DDX5/RNA pol II at the *WTIP* locus.

To test whether the knock down of *SUNO1* using siRNA (that displayed specific phenotype) compromises SUNO1 transcription, we determined RNA pol II occupancy on the *SUNO1* gene body in control and *SUNO1* siRNA-treated (siRNA targeting the 3’end of the gene). Recent study from Mendell laboratory indicated that antisense oligos targeting the 3’end of the gene does not compromise the transcription of the gene of interest, but preferentially degrades only the RNA (Lee and Mendell, 2020). ChIP-qPCR revealed that RNA pol II showed similar occupancy in the body of *SUNO1* gene in control and *SUNO1*-depleted cells (Figure 5Fa). This result suggested that the phenotype caused by *SUNO1* depletion was not due to reduced transcription activity of *SUNO1* but due to the loss of *SUNO1* RNA. In addition, unlike the known unstable and poly A- eRNAs (enhancer RNAs), *SUNO1* is a relatively stable poly A^+^ RNA (Figure 2—figure supplement 2B and Figure 2—figure supplement 3F).

9) The WTIP studies are poorly described. Why were the microarray studies done using SUNO1 siRNA rather than SUNO1 KO cells?

We thank the reviewer for his/her comments. We have now provided more data supporting the role of SUNO1/WTIP axis in regulating the YAP activity (please see below for more details). We wanted to distinguish primary (directly) targets from secondary (probably caused by cell cycle arrest) targets of *SUNO1*, the expression of which are altered immediately upon *SUNO1* depletion, even before the occurrence of cell cycle defects. Therefore, a transient knockdown of *SUNO1* (an early as well as late time point) is better to serve the purpose than a stably knocked-out cell line. Similar assays have been done previously to identify direct target of lncRNAs (Bergmann et al., 2015).

What time point is the immunoblot (Figure 4C).

Immunoblots were performed 72 h post si*SUNO1* transfection.

There appears to be no description of siRNA sequences.

We sincerely apologize for not showing the siRNA sequences clearly. We have now included the positions as well as the sequences of the siRNAs (Figure 2—figure supplement 3A; Supplementary file 8) in the revised manuscript.

It appears that two siRNA were used in Figure 4—figure supplement 1D, but the sequences and time are not described.

We have now included this information in the figure legend section of the manuscript.

Only a single siRNA was used to deplete WTIP (Figure 4—figure supplement 1G).

The siWTIP we used in the manuscript is a SMARTpool siRNA (Dharmacon, L-023639-020005), which contains 4 independent siRNAs. In addition, we now performed knockdown experiments using another DsiRNA from IDT and observed similar cell cycle phenotype (Author response images 2 and 3).

**Author response image 2. sa2fig2:** RT-qPCR to detect the levels of WTIP and YAP1 targets in control and WTIP-depleted cells. Data are presented as Mean ± SD, n=3. Unpaired two-tail t-tests are performed. **p* < 0.05.

**Author response image 3. sa2fig3:** Flow cytometry analyses to quantify the G1, S and G2/M population of cells in control and WTIP-depleted cells. Histograms from one of the replicates are shown. Population of G1, S and G2/M cells are quantified by de novo FCS Express 5 software. Data are presented as Mean ± SD, n=3. Unpaired two-tail t-tests are performed. ns, not significant; **p* < 0.05, ***p* < 0.01, ****p* < 0.001.

2.2

The peak of SUNO1 expression is in S phase (Figure 2—figure supplement 1C) while WTIP peaks in G1/S (Figure 4—figure supplement 1F) – if SUNO1 increases WTIP expression, what accounts for this?

We apologize for not describing this part of the result clearly. Our cell cycle RNA-seq data revealed that similar to *SUNO1*, WTIP showed enhanced expression in both G1/S and S phases of the cell cycle (Figure 4—figure supplement 1E). Although *SUNO1* level peaks in early S phase, its level is significantly increased at the G1/S boundary (Figure 2—figure supplement 1C), suggesting that the induction of *SUNO1* happens before the cells enter S phase. The further upregulation of *SUNO1* level may be necessary for the maintenance of the relatively high level of WTIP during S phase. The differential peaks of *SUNO1* RNA and WTIP protein could be controlled by multiple factors, including differential stability of *SUNO1* RNA and WTIP protein. Similarly, the decrease of *SUNO1* level at late S phase and G2 phase is concomitant with the drop of WTIP RNA levels.

The PI FACS analysis of siWTIP (Supplementary Figure 4D) is not sufficient to support the conclusions presented (see point 3, above).

We observed cell cycle defect phenotype with two independent set of siRNAs against WTIP. In addition, we now performed BrdU-PI flow analyses in control and WTIP-depleted cells, and it again phenocopies cell cycle defect caused by *SUNO1*-depletion (increased G1 population with a concomitant decrease in S phase) (Figure 4—figure supplement 1H).

10) Figure 4C shows that siSUNO1 decreases YAP expression – what is the mechanism?

Recent studies have shown that *YAP1* positively autoregulates its own expression (Vazquez-Marin et al., 2019). We observed several TEAD4 binding sites on the YAP1 promoter (Author response image 4). Besides TEAD4, *YAP1* promoter also contains FOS binding sites, and YAP1 is also shown to positively regulate FOS activity. These observations support the model that *YAP1* gene expression could be positively regulated by YAP1/TEAD4 or FOS axes.

**Author response image 4. sa2fig4:** UCSC genome view of the promoter of *YAP1* gene. ChIP-seq data set revealed TEAD and JUND binding signature on the *YAP1* promoter.

This decrease could be sufficient to explain the loss of YAP transcriptional activity. Does SUNO1 affect YAP localization?

We thank this reviewer for his/her suggestion. We performed YAP1 immunofluorescence staining along with EdU incorporation assay in *SUNO1*-depleted cells (Figure 4D). We observed reduced levels, including the nuclear pool of YAP1 in the *SUNO1*-depleted cells, supporting our observations that *SUNO1* depletion reduced the RNA and protein levels of YAP1.

Is this caused by SUNO1 deficiency or WTIP deficiency – can be tested by complementation analysis?

To test whether cell cycle defects observed in *SUNO1*-depleted cells is due to reduced WTIP expression, we performed WTIP rescue experiments, where we generated a stable HCT116 cell line expressing Dox-inducible EGFP-WTIP and performed *SUNO1* knockdown with or without EGFP-WTIP overexpression (Figure 4—figure supplement 2A-C). WTIP overexpression rescued the defects in the YAP1 levels observed upon *SUNO1* depletion (Figure 4—figure supplement 2C). In addition, WTIP-overexpressed cells partially rescued the cell cycle defects observed upon *SUNO1* depletion (Figure 4—figure supplement 2B). The WTIP-overexpressed/*SUNO1*-depleted cells showed statistically significant decrease in G1 and increase in S phase population, implying partial rescue (please also see the comments for point 11).

11) The authors should strengthen the functional connection between SUNO1and WTIP/ YAP1. Does WTIP or YAP1 rescue SUNO1 mediated functions?

We thank the reviewer for their suggestion. We decided to focus on WTIP, since our data supports *SUNO1* to be promoting the transcription of *WTIP* during cell cycle. We have now data demonstrating that exogenously expressed WTIP could partially rescue the cell cycle as well as YAP1 levels observed upon *SUNO1* depletion. For this, we generated a Tet/ON EGFP-WTIP stable cell line (Figure 4—figure supplement 2A-C). The overexpression of WTIP partially rescued the S phase defect of *SUNO1*-depleted cells (Figure 4—figure supplement 2B) and the cellular levels of YAP1 (Figure 4—figure supplement 2C).

WTIP overexpression did not rescue the DNA damage phenotype associated with *SUNO1* depletion (Figure 4—figure supplement 2A), as these cells continued to show increased levels of p53. These results indicate that the DNA damage phenotype observed upon *SUNO1* depletion might not be due to defects in WTIP/YAP1 activity. *SUNO1* might in addition play independent roles in modulating the expression of genes controlling DNA damage pathway, which would be the focus of future investigations.

The significance of interaction between SUNO1 and DDX5 in target gene regulation is unclear if DDX5 recruitment to target gene promoters is not dependent on SUNO1. How does SUNO1 affects the interaction between DDX5 and RNA pol II?

We completely agree with the reviewer’s suggestion that it would be ideal to determine the mechanism by which *SUNO1* modulates the DDX5/RNA pol II chromatin interactions. Earlier studies reported that DDX5 facilitates the recruitment/stabilization of RNA pol II to the promoters of cell cycle genes (Mazurek et al., 2012). Several other studies have also demonstrated the involvement of DDX5 in regulating RNA pol II activity, though the mechanism by which DDX5 controls RNA pol II loading remained to be established (Clark et al., 2013; Rossow and Janknecht, 2003).

Based on our results, we propose that *SUNO1*-DDX5 RNP complex at *WTIP* promoter may either confer specificity in recruiting RNA pol II to *WTIP* promoter, and/or stimulate the transcriptional coactivator activity of DDX5. Earlier studies, demonstrating the role of the SRA lncRNA in promoting DDX5 activity, support such a model (Caretti et al., 2006). In addition, a recent study showed that the *CONCR* lncRNA interacts with another helicase, DDX11, and regulates its enzymatic activity (Marchese et al., 2016). We therefore speculate that the mode of action of *SUNO1* may represent a wider spread mechanism in which lncRNAs interact with DEAD box family DNA/RNA helicases to modulate their location and co-transcriptional activity.

However, experiments to determine the molecular mechanism will include several large-scale experiments, such as in vitro RNA: protein reconstitution experiments. Such experiments may take several months to a year to complete, especially considering the labs at UIUC are allowed to work only at 50% of its capacity due to Covid-19 safety measures. We therefore intend to take up this task for future investigations.

12) The authors should provide further evidence to strengthen the connection to Hippo signaling. For example, does SUNO1 deletion affect YAP cellular localization, in addition to its expression level?

We thank the reviewer for his/her suggestion. We have observed reduced cellular levels of YAP1 in *SUNO1*-depleted cells (Figure 4D). In addition, we also observed increased protein levels of active phospho-LATS1 kinase (phospho-LATS1 inhibits the nuclear import of YAP1) in *SUNO1*-depleted cells (Figure 4C). Active LATS1, by phosphorylating YAP1, inhibits the nuclear import of YAP1, ultimately resulting in YAP1 degradation.

Does it also affect the TAZ level?

Thanks for the suggestion. We observed decreased levels of TAZ in *SUNO1*-depleted cells. (Figure 4C).

Expression of other bona fide YAP/TAZ target genes, such as CYR61 and ANKRD1, should be included in the analysis.

In addition to CTGF (a canonical YAP1 target), we observed reduced levels of *CYR61* in *SUNO1*-depleted cells (Figure 4B). However, *SUNO1* depletion in HCT116 cells did not change the levels of ANKRD1. Several of the YAP1 targets are reported to be cell type specific. Also, ANKRD1 might be regulated by other factors in HCT116 cell to circumvent the influence of the decrease of YAP1 activity.

In addition, we also demonstrated defects in the YAP1/TEAD-mediated transcriptional induction of CTGF promoter-bearing reporter gene in *SUNO1*-depleted cells (Figure 4E).

In addition, they should examine whether SUNO1 affects phosphorylation of the upstream Hippo kinases, Mst and LATS.

We thank the reviewer for his/her suggestion. We observed an increase of pLATS1 in *SUNO1* depleted cells (Figure 4C). pLATS is an inhibitor of YAP1.

13) It was recently reported that expression of the components of the Hippo pathway, including YAP/TAZ, oscillate during mitotic cell cycle (Kim et al., 2019). Thus, it is important to test whether cell cycle regulation of YAP/TAZ is indeed dependent on SUNO1 or WTIP in this context.

We thank this reviewer for this comment. Kim et al. manuscript revealed that YAP levels are increased during G1/S and S phase of the cell cycle. Their data is consistent with our observations that *SUNO1* promotes the expression of YAP1 by positively regulating *WTIP* transcription. Studies have demonstrated that YAP1 promotes S phase entry by positively regulating the expression of genes, controlling replication and S phase entry (Shen and Stanger, 2015). Reduced levels of WTIP, YAP1, YAP1 targets such as CTGF, Cyclin D1 and increased levels of the inhibitor of YAP1, pLATS1 in *SUNO1*-depleted cells further support the model that *SUNO1* regulates the S phase expression of YAP1, potentially by promoting the expression of *WTIP* in *cis*.

14) The authors state that "Interestingly, a major fraction of these genes (71%), including WTIP, showed a positive correlation in expression with SUNO1 across colon cancer patients" However the statistical significance of the correlated genes is not provided.

All of the reported positively correlated genes with *SUNO1* exhibited a p-value < 0.01 at a 5% FDR. We have now included this information both in the figure legend and manuscript text to provide statistical significance of our observations.

15) The soft agar assays cannot be interpreted without studies of 2D cultures. Is the loss of growth in agar simply because the SUNO1 KO cells do not grow?

We sincerely apologize for the confusion. In the soft agar anchorage independent assay (Figure 6D), we seeded equal number of control and *SUNO1*-KO cells. Colony numbers were then counted regardless of the size of individual colonies. In this way, we minimize the effect caused by different growth rates of cells. In this sense, *SUNO1* KO cells showed reduced colony number (because of *SUNO1*’s role in cell proliferation) as well as colony size (its involvement in anchorage-independent growth). In addition, we have performed 2D growth curve analyses in WT and *SUNO1* KO cells. The *SUNO1* KO cells showed a much slower growth compared to the WT control cells (Figure 3—figure supplement 1H). Based on these results, we conclude that *SUNO1* promotes cell proliferation as well as tumor growth (the ability of tumor cells to grow in an anchorage-independent way).

16) Many experiments use a single siRNA. At least two siRNAs or siRNA/CRISPR strategies should be used to strengthen their conclusions. In addition to qRT-PCR validation of target genes, Western blot analyses should be included.

We thank this reviewer for his/her suggestions. We have now performed most of the experiments using two independent siRNAs. In addition, similar phenotypes were also observed in the *SUNO1* KO cells.

In addition to RT-qPCR validation, levels of several proteins that are part of YAP/Hippo signaling (YAP, pLATS, CTGF, Cyclin D, p15/PAF) were also assessed in control and *SUNO1*-depleted cells by immunoblotting (Figure 4C and Figure 4—figure supplement 1B).

References:

Andrysik, Z., Galbraith, M. D., Guarnieri, A. L., Zaccara, S., Sullivan, K. D., Pandey, A.,… Espinosa, J. M. (2017). Identification of a core TP53 transcriptional program with highly distributed tumor suppressive activity. Genome Res, 27(10), 1645-1657. doi:10.1101/gr.220533.117

Bergmann, J. H., Li, J., Eckersley-Maslin, M. A., Rigo, F., Freier, S. M., and Spector, D. L. (2015). Regulation of the ESC transcriptome by nuclear long noncoding RNAs. Genome Res, 25(9), 1336-1346. doi:10.1101/gr.189027.114

Caretti, G., Schiltz, R. L., Dilworth, F. J., Di Padova, M., Zhao, P., Ogryzko, V.,… Sartorelli, V. (2006). The RNA helicases p68/p72 and the noncoding RNA SRA are coregulators of MyoD and skeletal muscle differentiation. Dev Cell, 11(4), 547-560. doi:10.1016/j.devcel.2006.08.003

Dimitrova, N., Zamudio, J. R., Jong, R. M., Soukup, D., Resnick, R., Sarma, K.,... Jacks, T. (2014). LincRNA-p21 activates p21 in *cis* to promote Polycomb target gene expression and to enforce the G1/S checkpoint. Mol Cell, 54(5), 777-790. doi:10.1016/j.molcel.2014.04.025

Lee, J. S., & Mendell, J. T. (2020). Antisense-Mediated Transcript Knockdown Triggers Premature Transcription Termination. Mol Cell, 77(5), 1044-1054 e1043. doi:10.1016/j.molcel.2019.12.011

Liu, Y., Chen, S., Wang, S., Soares, F., Fischer, M., Meng, F.,… He, H. H. (2017). Transcriptional landscape of the human cell cycle. Proc Natl Acad Sci U S A, 114(13), 3473-3478. doi:10.1073/pnas.1617636114

Marchese, F. P., Grossi, E., Marin-Bejar, O., Bharti, S. K., Raimondi, I., Gonzalez, J.,… Huarte, M. (2016). A Long Noncoding RNA Regulates Sister Chromatid Cohesion. Mol Cell, 63(3), 397-407. doi:10.1016/j.molcel.2016.06.031

Mazurek, A., Luo, W., Krasnitz, A., Hicks, J., Powers, R. S., & Stillman, B. (2012). DDX5 regulates DNA replication and is required for cell proliferation in a subset of breast cancer cells. Cancer Discov, 2(9), 812-825. doi:10.1158/2159-8290.CD-12-0116

Prasanth, S. G., Prasanth, K. V., and Stillman, B. (2002). Orc6 involved in DNA replication, chromosome segregation, and cytokinesis. Science, 297(5583), 1026-1031. doi:10.1126/science.1072802

Rao, S. S. P., Huang, S. C., Glenn St Hilaire, B., Engreitz, J. M., Perez, E. M., Kieffer-Kwon, K. R.,… Aiden, E. L. (2017). Cohesin Loss Eliminates All Loop Domains. Cell, 171(2), 305-320 e324. doi:10.1016/j.cell.2017.09.026

Rossow, K. L., & Janknecht, R. (2003). Synergism between p68 RNA helicase and the transcriptional coactivators CBP and p300. Oncogene, 22(1), 151-156. doi:10.1038/sj.onc.1206067

Seiler, J., Breinig, M., Caudron-Herger, M., Polycarpou-Schwarz, M., Boutros, M., and Diederichs, S. (2017). The lncRNA VELUCT strongly regulates viability of lung cancer cells despite its extremely low abundance. Nucleic Acids Res, 45(9), 5458-5469. doi:10.1093/nar/gkx076

Shen, Z., Sathyan, K. M., Geng, Y., Zheng, R., Chakraborty, A., Freeman, B.,... Prasanth, S. G. (2010). A WD-repeat protein stabilizes ORC binding to chromatin. Mol Cell, 40(1), 99-111. doi:10.1016/j.molcel.2010.09.021

Shen, Z., and Stanger, B. Z. (2015). YAP regulates S-phase entry in endothelial cells. PLoS One, 10(1), e0117522. doi:10.1371/journal.pone.0117522

Varley, K. E., Gertz, J., Roberts, B. S., Davis, N. S., Bowling, K. M., Kirby, M. K.,… Myers, R. M. (2014). Recurrent read-through fusion transcripts in breast cancer. Breast Cancer Res Treat, 146(2), 287297. doi:10.1007/s10549-014-3019-2

Vazquez-Marin, J., Gutierrez-Triana, J. A., Almuedo-Castillo, M., Buono, L., Gomez-Skarmeta, J. L., Mateo, J. L.,... Martinez-Morales, J. R. (2019). yap1b, a divergent Yap/Taz family member, cooperates with yap1 in survival and morphogenesis via common transcriptional targets. Development, 146(13). doi:10.1242/dev.173286

Yekezare, M., Gomez-Gonzalez, B., and Diffley, J. F. (2013). Controlling DNA replication origins in response to DNA damage – inhibit globally, activate locally. J Cell Sci, 126(Pt 6), 1297-1306. doi:10.1242/jcs.096701